# Lipolysis of bone marrow adipocytes is required to fuel bone and the marrow niche during energy deficits

Ziru Li[1], Emily Bowers[2], Junxiong Zhu[3,4], Hui Yu[1], Julie Hardij[1], Devika P Bagchi[1], Hiroyuki Mori[1], Kenneth T Lewis[1], Katrina Granger[1], Rebecca L Schill[1], Steven M Romanelli[1], Simin Abrishami[2], Kurt D Hankenson[3], Kanakadurga Singer[1,2], Clifford J Rosen[5], Ormond A MacDougald[1,6]*

[1]University of Michigan Medical School, Department of Molecular & Integrative Physiology, Ann Arbor, United States; [2]University of Michigan Medical School, Department of Pediatrics, Ann Arbor, United States; [3]Department of Orthopedic Surgery, University of Michigan Medical School, Ann Arbor, United States; [4]Department of Orthopedic Surgery, The Second Affiliated Hospital, Zhejiang University School of Medicine, Hangzhou, China; [5]Maine Medical Center Research Institute, Scarborough, United States; [6]University of Michigan Medical School, Department of Internal Medicine, Ann Arbor, United States

**Summary** To investigate roles for bone marrow adipocyte (BMAd) lipolysis in bone homeostasis, we created a BMAd-specific Cre mouse model in which we knocked out adipose triglyceride lipase (ATGL, *Pnpla2* gene). BMAd-*Pnpla2*⁻/⁻ mice have impaired BMAd lipolysis, and increased size and number of BMAds at baseline. Although energy from BMAd lipid stores is largely dispensable when mice are fed ad libitum, BMAd lipolysis is necessary to maintain myelopoiesis and bone mass under caloric restriction. BMAd-specific *Pnpla2* deficiency compounds the effects of caloric restriction on loss of trabecular bone in male mice, likely due to impaired osteoblast expression of collagen genes and reduced osteoid synthesis. RNA sequencing analysis of bone marrow adipose tissue reveals that caloric restriction induces dramatic elevations in extracellular matrix organization and skeletal development genes, and energy from BMAd is required for these adaptations. BMAd-derived energy supply is also required for bone regeneration upon injury, and maintenance of bone mass with cold exposure.

*For correspondence:
macdouga@umich.edu

Competing interest: The authors declare that no competing interests exist.

## Editor's evaluation

Through the cell selective deletion in bone marrow marrow adipocytes (BMAds) of an enzyme for lipolysis, the authors demonstrate elegantly a role for this pathway in the maintenance of hematopoiesis and bone mass during periods of caloric stress, injury and energy restriction. This study establishes for the first time a role for the bone marrow adipocyte in energy balance.

## Introduction

Adipocytes are found throughout the body, and can be classified generally into white, brown, beige/BRITE and bone marrow adipose tissues (BMAT); however, many other niche depots exist, and adipocyte subpopulations within specific depots have recently been identified (*Bagchi et al., 2018*; *Emont et al., 2022*; *Sárvári et al., 2021*). In an adult human, 50–70% of the bone marrow cavity is filled with BMAT, which contributes ~10% of the total body fat mass (*Cawthorn et al., 2014*). BMAT was

identified in the 19th century, and has been assumed to play significant roles in local bone homeostasis and hematopoiesis (*Li et al., 2018*; *Li and MacDougald, 2021*). Clinical associations generally demonstrate inverse relationships between BMAT and bone mass (*Shen et al., 2007*), or BMAT and circulating immune cells (*Polineni et al., 2020*), which may be due to the interactions between cells within the bone marrow niche. In addition to bone marrow adipocytes (BMAds), the bone marrow niche contains osteoblasts, osteoclasts, hematopoietic cells, stromal/mesenchymal cells, blood vessels, and nerves (*Vogler and Murphy, 1988*). The relationships between BMAds, bone cells, and hematopoietic cells are influenced by their shared location within bone, an anatomically restricted system, such that expansion of one cell type is by necessity at the expense of others. For example, elevated BMAT is inversely correlated with low bone mass of aging and diabetes, whereas expansion of BMAT is associated with multiple hematopoietic disorders (*Li and MacDougald, 2021*). Mechanistic links underlying these associations have proven challenging to investigate because these often involve complex intercellular, endocrine, and/or central mechanisms.

In addition to serving as an energy source, BMAds potentially influence the marrow niche through cell-to-cell contact, release of extracellular vesicles, and secretion of adipokines (e.g. adiponectin and leptin) and cytokines (e.g. adipsin, RANK ligand, and stem cell factor; *Aaron et al., 2021*; *Li et al., 2018*). Removal of these stimuli in mouse models of lipodystrophy, which lack BMAT, results in increased bone mass (*Corsa et al., 2021*; *Zou et al., 2020*; *Zou et al., 2019*). However, use of these models to investigate direct effects of BMAT depletion is confounded by concurrent loss of white and brown adipose depots, which also regulate bone mass through myriad mechanisms, including secretion of adipokines (*Riddle and Clemens, 2017*; *Zou et al., 2019*). Although loss of BMAT in lipodystrophic mice was integral to the original finding that BMAT is a negative regulator of hematopoiesis, positive effects of BMAT on hematopoiesis have also been observed; these differences are attributed to use of distinct animal models and analysis of different skeletal sites (*Ambrosi et al., 2017*; *Naveiras et al., 2009*; *Zhou et al., 2017*). In rodents, there are two readily identifiable BMAd populations: constitutive BMAT (cBMAT) and regulated BMAT (rBMAT) (*Scheller et al., 2015*). cBMAT typically exists in distal tibia and caudal vertebrae, appears early in life, and has the histological appearance of white adipose tissue. rBMAT is found in proximal tibia and distal femur, and is comprised of single or clustered BMAds interspersed with hematopoietic cells. Although cBMAT generally resists change in response to altered physiological states, rBMAT expands with aging, obesity, diabetes, caloric restriction (CR), irradiation, and estrogen deficiency, and is reduced by cold exposure, β3-agonist, exercise, and vertical sleeve gastrectomy (*Li et al., 2018*; *Li et al., 2019*). These treatments also cause alterations to the skeleton and/or formation of blood cells, some of which may be secondary to effects on BMAds.

In contrast, BMAds have also been thought to fuel maintenance of bone and hematopoietic cellularity because of their shared physical location in the marrow niche. Indeed, *Steele, 1884* wrote that fat in marrow 'nourishes the skeleton'; however, that conjecture has not been formally tested because current methods to target BMAds lack penetrance or cause recombination in other cell types, such as white adipocytes, osteoblasts, or bone marrow stromal cells, complicating the interpretation of interactions between BMAds and cells of the marrow niche. To circumvent this problem, we created a BMAd-specific Cre mouse model based on expression patterns of endogenous *Osterix* and *Adipoq*, and investigated the role of BMAds as a local energy source by deleting ATGL/*Pnpla2*, the rate-limiting enzyme of lipolysis. Consistent with *Pnpla2* deficiency in white and brown adipocytes (*Ahmadian et al., 2011*), BMAd-*Pnpla2*[-/-] mice have impaired BMAd lipolysis, resulting in hypertrophy and hyperplasia of BMAT. Despite significant increases in bone marrow adiposity, hematopoietic abnormalities of BMAd-*Pnpla2*[-/-] mice are negligible under basal conditions. However, the recovery of bone marrow myeloid lineages following sublethal irradiation is impaired with CR, and is further reduced by BMAd-*Pnpla2* deficiency. Similarly, the proliferative capacity of myeloid progenitors is also decreased with CR, and further inhibited in mice lacking BMAd-*Pnpla2*. Whereas alterations in bone parameters were not observed in ad libitum fed BMAd-*Pnpla2*[-/-] mice, bone loss occurred under conditions of elevated energy needs such as bone regeneration or cold exposure, or reduced energy availability such as CR. Reduction of bone mass in CR BMAd-*Pnpla2*[-/-] mice is likely due to impaired osteoblast functions such as expression of extracellular matrix genes and creation of osteoid. Gene profiling reveals that *Pnpla2* deletion largely blocks CR-induced genes within pathways of extracellular matrix organization and skeletal development,

indicating that BMAd-derived energy contributes to skeletal homeostasis under conditions of negative energy balance.

## Results

### Generation of a BMAd-specific Cre mouse model (BMAd-Cre)

Based on previous studies showing that *Osterix* traces to osteoblasts and BMAds, but not to white adipocytes (*Chen and Long, 2013*; *Mizoguchi et al., 2014*), we used CRISPR/Cas9 to create *Osterix-FLPo* mice with an in-frame fusion of *Osterix* and optimized *FLPo*, separated by a *P2A* self-cleaving sequence to allow independent functioning of the two proteins (*Figure 1A*). To validate tissue-specific expression and FLPo efficiency, we bred *Osterix-FLPo* mice to FLP-dependent EGFP reporter mice, and observed EGFP-positive osteocytes, osteoblasts, BMAds, and a subset of marrow stromal cells within the bone (*Figure 1B and C*).

We next created FLPo-dependent *Adipoq-Cre* (FAC) mice, which contain an internal ribosome entry sequence (*IRES*) followed by FLPo-dependent *Cre* in reverse orientation within the 3'-untranslated region (UTR) of endogenous *Adipoq* gene (*Figure 1D*). FLPo expressed from the *Osterix* locus recombines *Cre* to the correct orientation in progenitors of osteoblasts and BMAds. However, because *Adipoq* is selectively expressed in adipocytes (*Eguchi et al., 2011*), *Cre* was expressed in BMAds, but not in osteoblasts or other adipose depots. Consistent with this schema, *Cre* in the correct orientation (flipped band, *Figure 1—figure supplement 1A–C*) was only observed in caudal vertebral DNA of mice that were positive for at least one copy each of *Osterix* (Mut band) and FAC (Ori band). The correct orientation of *Cre* (flipped band) was also observed in mRNA isolated from distal tibiae and caudal vertebrae, but not WAT depots, of *Osterix-FLPo* mice positive for FAC (*Figure 1—figure supplement 1D*). The correct insertion of the sequence in *Adipoq* 3'-UTR was validated by Sanger sequencing following genomic PCRs that spanned endogenous *Adipoq* sequences, homology arms, *IRES* and *Cre* (*Figure 1—figure supplement 1E*). To investigate the cell type specificity of Cre activity, we next bred BMAd-*Cre* mice to mT/mG reporter mice (*Muzumdar et al., 2007*; *Figure 1E*), in which all tissues and cells express red fluorescence (membrane-targeted tdTomato; mT) at baseline, and will express membrane-targeted EGFP in the presence of

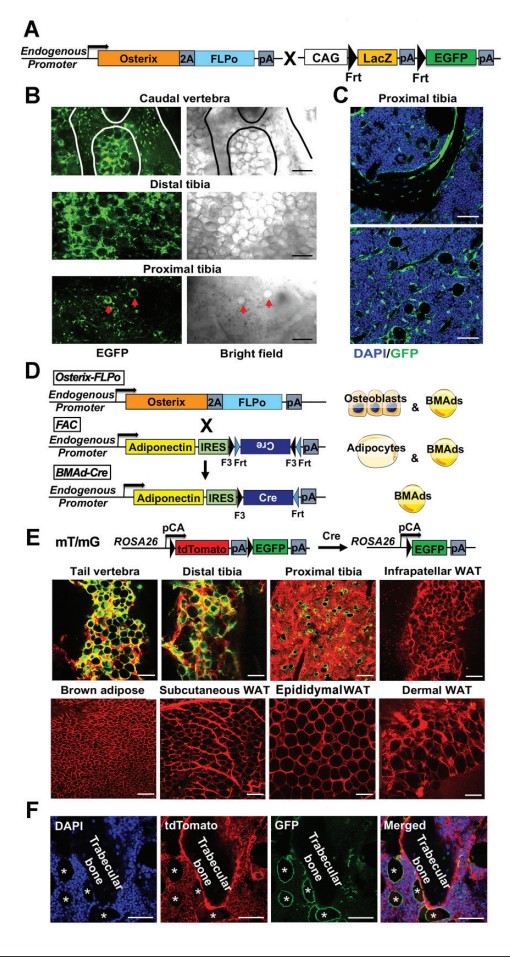

**Figure 1.** Generation of a BMAd-specific Cre mouse model (BMAd-*Cre*). (**A**) Efficacy of *Osterix-FLPo* was evaluated by crossing with FLP-dependent EGFP reporter to yield *Osterix*-EGFP. (**B**) *Osterix*-EGFP male mice at 16 weeks were sacrificed. Fresh tissue confocal microscopy was performed on bisected caudal vertebrae, distal tibia and proximal tibia. Red arrows indicate singly dispersed BMAds. Scale bar; 100 µm. (**C**) Frozen-sections of proximal tibial slides from *Osterix*-EGFP were stained with Anti-GFP (green) and DAPI (blue). Scale bar; 50 µm. (**D**) Schematic of how *Osterix-FLPo* recombines FLPo-activated *Adipoq-Cre* (FAC) in BMAd-*Cre* mice to restrict expression of *Cre* to BMAds. (**E**) BMAd-*Cre* mice were bred with mT/mG reporter mice and the resulting BMAd-mT/mG mice were sacrificed at 16 weeks of age. Cellular fluorescence was evaluated by fresh tissue confocal microscopy. Scale bar; 100 µm. (**F**) Proximal tibial sections from BMAd-mT/mG mice were stained with antibodies to tdTomato (red) and EGFP (green), and nuclei were counterstained with DAPI (blue). * indicates BMAds. Scale bar; 50 µm.

The online version of this article includes the following source data and figure supplement(s) for figure 1:

**Figure supplement 1.** Genotyping strategy and validation of BMAd-*Cre*.

*Figure 1 continued on next page*

*Figure 1 continued*

**Figure supplement 2.** Insertion of *IRES-Cre* cassette in the 3'UTR of endogenous *Adipoq* decreases expression of adiponectin but does not cause a metabolic or bone phenotype.

**Figure supplement 2—source data 1.** Insertion of *IRES-Cre* cassette in the 3'UTR of endogenous *Adipoq* decreases expression of adiponectin but does not cause a metabolic or bone phenotype.

cell-specific Cre. We observed loss of tdTomato and gain of EGFP in BMAT depots of caudal vertebrae (tail) and tibiae, but not in brown or white adipose depots, or other tissues/organs such as liver, pancreas, muscle and spleen (*Figure 1E* and *Figure 1—figure supplement 1F*). To evaluate conditions optimal for Cre-induced recombination, we visualized conversion of tdTomato to EGFP in BMAd-*Cre* mice at various ages and with different FAC copy numbers (*Figure 1—figure supplement 1G–H*) and found that the proportion of EGFP-positive BMAds increased with age and number of FAC alleles. Indeed, Cre efficiency was ~80% in both male and female mice over 16 weeks of age with one *Cre* allele, and over 90% at 12 weeks of age in mice with two *Cre* alleles (*Figure 1—figure supplement 1I*).

Previous studies found that a randomly inserted *Adipoq-Cre* bacterial artificial chromosome (BAC) causes recombination in osteoblasts (*Bozec et al., 2013*; *Eguchi et al., 2011*; *Mukohira et al., 2019*); thus, we tested whether this observation is true in BMAd-*Cre* mice, which rely on endogenous *Adipoq* expression that is further restricted by *Osterix*. When BMAd-*Cre* mice were bred to mT/mG reporter mice, only tdTomato-positive cells were detectable on trabecular bone surfaces (*Figure 1F*), suggesting that osteoblasts were not targeted. However, labeling of a small subset of stromal/dendritic cells was consistently observed, likely due to expression of *Adipoq* within this cell population (*Mukohira et al., 2019*).

Although insertion of the FLPo-activated *Cre* cassette into the 3′-UTR of endogenous *Adipoq* overcomes potential problems that arise from random genomic integration, we considered the possibility that placement of the *IRES-Cre* cassette within the 3′-UTR might alter the expression and/or secretion of endogenous adiponectin. We found that insertion of the *IRES-Cre* cassette caused a small, non-significant decrease in *Adipoq* mRNA expression in whole caudal vertebrae or distal tibiae, whereas adiponectin mRNA in white adipose tissue was elevated by more than threefold (*Figure 1—figure supplement 2A*). Thus, the cassette itself did not limit expression of mRNA. Instead, it appeared that the 2 kb *IRES-Cre* cassette within the 3'-UTR impairs translatability of the mRNA since expression of adiponectin protein in WAT and BMAT was decreased by ~50% (*Figure 1—figure supplement 2B and C*). Circulating adiponectin concentrations were decreased even further (*Figure 1—figure supplement 2D–F*), suggesting that flux of translated adiponectin protein was impaired through the secretion pathway. Of note, hypoadiponectinemia is associated with insulin resistance (*Cook and Semple, 2010*; *Li et al., 2021*), with discrepant effects reported on osteogenesis (*Lewis et al., 2021*). Thus, we evaluated systemic metabolism in BMAd-Cre mice and found that body weight, glucose tolerance, WAT depot weights, and tibial trabecular and cortical bone variables were unaffected by this degree of hypoadiponectinemia (*Figure 1—figure supplement 2G–P*). To minimize variability between treatments, all mice used in the following studies were positive for both *Osterix-FLPo* (predominantly two alleles) and FAC (mostly two alleles). Gene knockout mice and their controls were determined by the presence or absence of floxed alleles, respectively.

## Ablation of adipose triglyceride lipase (ATGL, *Pnpla2*) causes BMAT expansion

A fundamental function of adipocytes is to store excess energy as triacylglycerols and to release non-esterified fatty acids (NEFA) and glycerol during times of negative energy balance. ATGL is the first and rate-limiting enzyme in the lipolytic process; thus, to determine the physiological functions of BMAd lipolysis in bone metabolism and hematopoiesis, we generated BMAd-*Pnpla2*−/− mice, in which the gene encoding ATGL, *Pnpla2*, was specifically deleted in BMAds. We validated the deletion of ATGL in caudal vertebrae, because cBMAT is abundant in this location. Mutant *Pnpla2* mRNA was observed in caudal vertebrae of BMAd-*Pnpla2* mice, but not in WAT (*Figure 2A*). Possible sources for the remaining wildtype (WT) *Pnpla2* signal include periosteal WAT, non-adipocyte cells, or perhaps from incomplete deletion of *Pnpla2* in BMAds. Immunofluorescent staining confirmed loss of ATGL in proximal tibial rBMAds of BMAd-*Pnpla2*−/− mice (*Figure 2B*). ATGL protein was reduced substantially

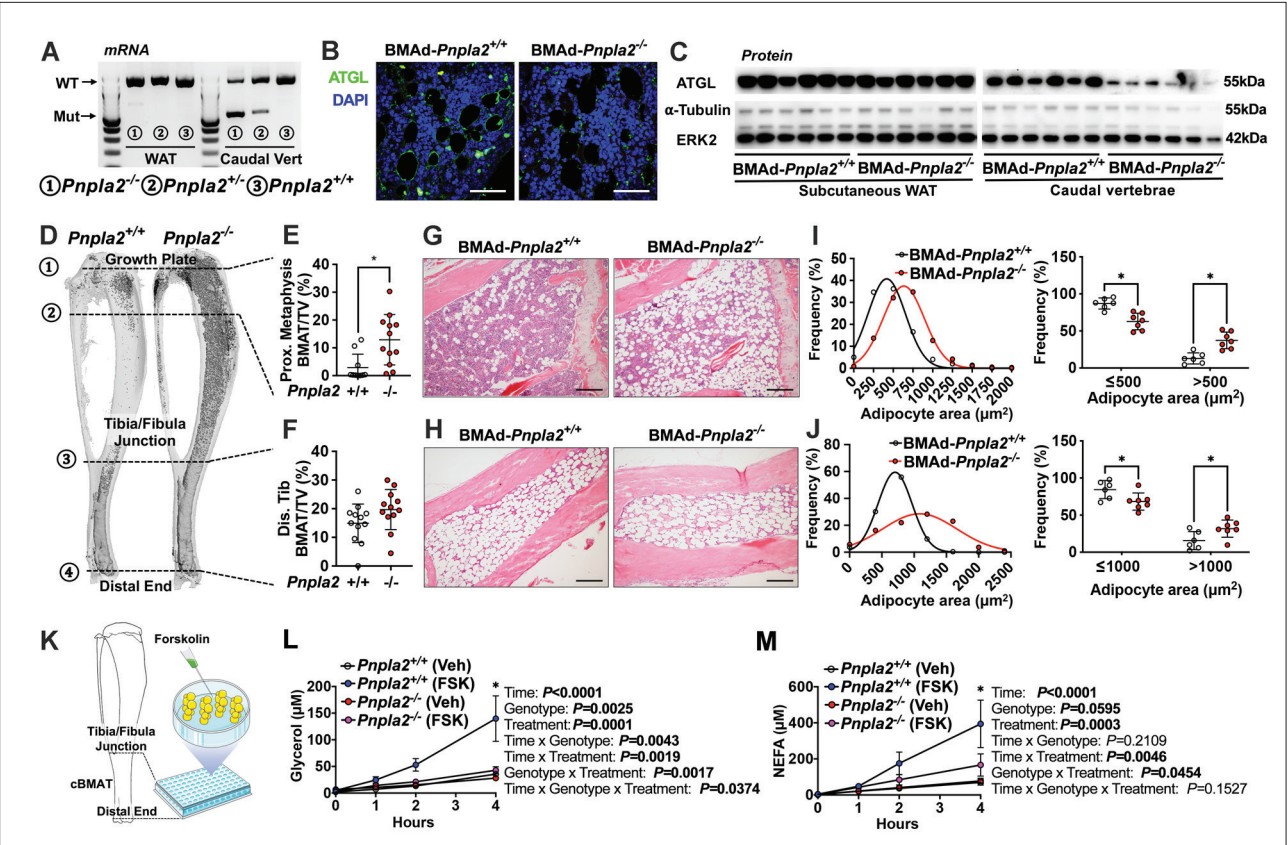

**Figure 2.** Ablation of adipose triglyceride lipase (ATGL; gene name *Pnpla2*) increases size and number of BMAd. (**A-J**) Male mice of the indicated genotypes at 24 weeks of age were euthanized for investigation of white adipose tissue (WAT) and bone. (**A**) RNA was extracted from WAT and caudal vertebrae, and converted to cDNA. PCR products for wildtype (WT; 1257 bp) *Pnpla2* and exon 2–7 knockout (Mut; 553 bp) bands were visualized. (**B**) Decalcified proximal tibiae were sectioned and used for immunofluorescent staining for ATGL (green) expression. Slides were counterstained with DAPI (blue) for nuclei. Scale bar; 50 µm. (**C**) Immunoblot analyses of ATGL, α-tubulin, and ERK2 in lysates from subcutaneous WAT and caudal vertebral. (**D-F**) Decalcified tibiae were stained with osmium tetroxide and visualized by µCT (**D**). BMAT in proximal (**E**) and distal (**F**) tibia was quantified. Data are expressed as mean ± SD. * indicates p<0.05 with a two-sample *t*-test. (**G–H**) Decalcified tibiae were paraffin-sectioned and stained with Hematoxylin & Eosin. Representative pictures were taken from proximal (**G**) and distal (**H**) tibia. Scale bar; 200 µm. (**I-J**) BMAd sizes from proximal (**I**) and distal (**J**) tibiae were quantified with MetaMorph software. Data are expressed as mean ± SD. * indicates p<0.05 with two-way ANOVA with Sidak's multiple comparisons test. (**K–M**) Distal tibial BMAT was flushed from female BMAd-*Pnpla2*[-/-] and their wildtype littermates at 24 weeks of age. For each n, distal tibial explants from two mice were combined per well and cultured in 2% BSA-HBSS solution (**K**). Subgroups from each genotype were treated with forskolin (FSK, 5 µM) or vehicle (Veh, DMSO). Glycerol (**L**) and non-esterified fatty acid (NEFA; **M**) in culture media at indicated time points were measured by colorimetric assay (n=3–4 per treatment). * indicates *Pnpla2*[+/+] (FSK) different from *Pnpla2*[+/+] (Veh) and from *Pnpla2*[-/-] (FSK) with p<0.05 with three-way ANOVA followed by Sidak's multiple comparisons test.

The online version of this article includes the following source data for figure 2:

**Source data 1.** Ablation of adipose triglyceride lipase (ATGL; gene name *Pnpla2*) increases size and number of BMAd.

in caudal vertebrae, but not in subcutaneous WAT (*Figure 2C*), which further confirmed specificity of BMAd-*Cre* recombinase for BMAT.

To quantify effects of *Pnpla2*-deficiency on bone marrow adiposity and cellularity, we used osmium tetroxide, and hematoxylin and eosin staining, to evaluate BMAT quantity and cellular details, respectively. We found that BMAT volume was significantly increased in proximal tibiae and throughout the endocortical compartment of BMAd-*Pnpla2*[-/-] mice, whereas differences in BMAT volume were not observed in distal tibiae, which was almost completely occupied by BMAT in control mice (*Figure 2D–F*). Histological analyses revealed increased BMAd number in proximal tibiae of BMAd-*Pnpla2*[-/-] mice (*Figure 2G and H*). Depletion of *Pnpla2* also caused BMAd hypertrophy in both proximal and distal tibia, with an increased proportion of BMAds larger than 500 µm² observed in proximal tibia, and larger than 1000 µm² observed in distal tibia (*Figure 2I and J*). Secretion of basal glycerol

and NEFA from ex vivo cultured explants of distal tibial BMAT was not different between genotypes. On the other hand, forskolin treatment greatly increased lipolysis in BMAT explants from control mice, but secretion of glycerol and NEFA from BMAd-*Pnpla2*⁻/⁻ explants remained unchanged from basal rates (*Figure 2K–M*). These results confirm that BMAds of BMAd-*Pnpla2*⁻/⁻ mice have impaired lipolysis.

## BMAd lipolysis is required to maintain bone homeostasis in male mice under conditions of CR, but not when mice are fed ad libitum

We next evaluated whether loss of BMAd lipolysis in male BMAd-*Pnpla2*⁻/⁻ mice is sufficient to influence systemic physiology, or function of bone cells within the marrow niche. In mice fed normal chow ad libitum, we did not observe differences in body weight, glucose tolerance, or weights of soft tissue, including subcutaneous WAT (sWAT), epididymal WAT (eWAT), and liver (*Figure 3—figure supplement 1A-E*), which further confirmed the tissue-specificity of our BMAd-*Pnpla2* knockout model. As observed above, *Pnpla2* deficiency caused expansion of proximal tibial rBMAT (*Figure 3—figure supplement 1F and G*). Interestingly, when dietary energy was readily available, trabecular bone volume fraction, bone mineral density and trabecular number tended to be lower in BMAd-*Pnpla2*⁻/⁻ mice, but statistical differences were not observed (*Figure 3A–C*).

Our experiments in ad libitum fed mice suggest that energy released from BMAds is dispensable for bone cell functioning when dietary energy is plentiful, which is supported by the comparable concentrations of glycerol and NEFA in circulation and in bone marrow supernatant in BMAd-*Pnpla2*⁺/⁺ and BMAd-*Pnpla2*⁻/⁻ mice (*Figure 3—figure supplement 1H and I*). To determine whether male BMAd-*Pnpla2*⁻/⁻ mice have impaired bone homeostasis when dietary energy is limited, we next challenged mice with a 30% CR for 6 weeks. BMAd-*Pnpla2*⁻/⁻ mice did not have altered body weight, glucose tolerance, or tissue weights comparing with their controls (*Figure 3—figure supplement 1J-N*), suggesting that BMAds were not a critical source of circulating energy under these conditions. Although *Pnpla2*⁻/⁻ mice fed ad libitum had increased rBMAT compared to controls (*Figure 3—figure supplement 1F and G*), CR stimulated rBMAT expansion in control mice such that BMAT volume and BMAd size were comparable between genotypes (*Figure 3—figure supplement 1O-S*). Despite having a similar amount of rBMAT, BMAd-*Pnpla2*⁻/⁻ mice fed CR diet had reduced trabecular bone volume fraction, connective density, and bone mineral density when compared to controls (*Figure 3D and E*). This reduction in bone mass is due to decreased trabecular bone number and increased trabecular spacing, without effects on trabecular thickness (*Figure 3F*). BMAds are a source of NEFA for vicinal osteoblasts (*Maridas et al., 2019*); thus, the reduction in trabecular bone in CR BMAd-*Pnpla2*⁻/⁻ mice likely resulted from impaired osteoblast function due to insufficient energy available in the bone marrow niche. Consistent with this notion, we observed a trend (p=0.07) towards reduced NEFA concentrations in bone marrow supernatant of CR BMAd-*Pnpla2*⁻/⁻ mice, whilst circulating glycerol and NEFA concentrations were not different between genotypes (*Figure 3—figure supplement 1T and U*). Although FAC mice have hypoadiponectinemia at baseline, circulating adiponectin was still increased dramatically in response to CR. Comparable adiponectin concentrations between BMAd-*Pnpla2*⁺/⁺ and BMAd-*Pnpla2*⁻/⁻ mice (*Figure 3—figure supplement 1V*) excludes the possibility that differences in circulating adiponectin may confound the bone phenotype in BMAd-*Pnpla2*⁻/⁻ mice. Finally, changes in cortical bone area and thickness were not observed in BMAd-*Pnpla2*⁻/⁻ mice with either ad libitum or CR diet (*Figure 3—figure supplement 2A-D*), perhaps because cortical bone surfaces are less active.

To investigate mechanisms underlying trabecular bone loss in CR BMAd-*Pnpla2*⁻/⁻ mice, we next measured circulating markers of bone cell activity and performed histomorphometry. Osteoblast and osteoclast numbers and osteoclast surface were not changed by BMAd-*Pnpla2* deficiency under ad libitum conditions (*Figure 3—figure supplement 2E and F*). Although circulating markers of bone formation (P1NP) and osteoclast activation (RANK Ligand) were not altered in BMAd-*Pnpla2*⁻/⁻ mice, markers for bone resorption (CTX-1) and osteoclast activation (TRACP5b) were decreased or showed a strong trend towards reduction, respectively (*Figure 3—figure supplement 2G-J*). Although with CR, osteoblast numbers were not altered by a deficiency of BMAT lipolysis, the bone formation marker P1NP was decreased (*Figure 3G and I*), suggesting that osteoblast functions were impaired. Osteoclast numbers were increased in CR BMAd-*Pnpla2*⁻/⁻ mice, but the osteoclast surface per bone

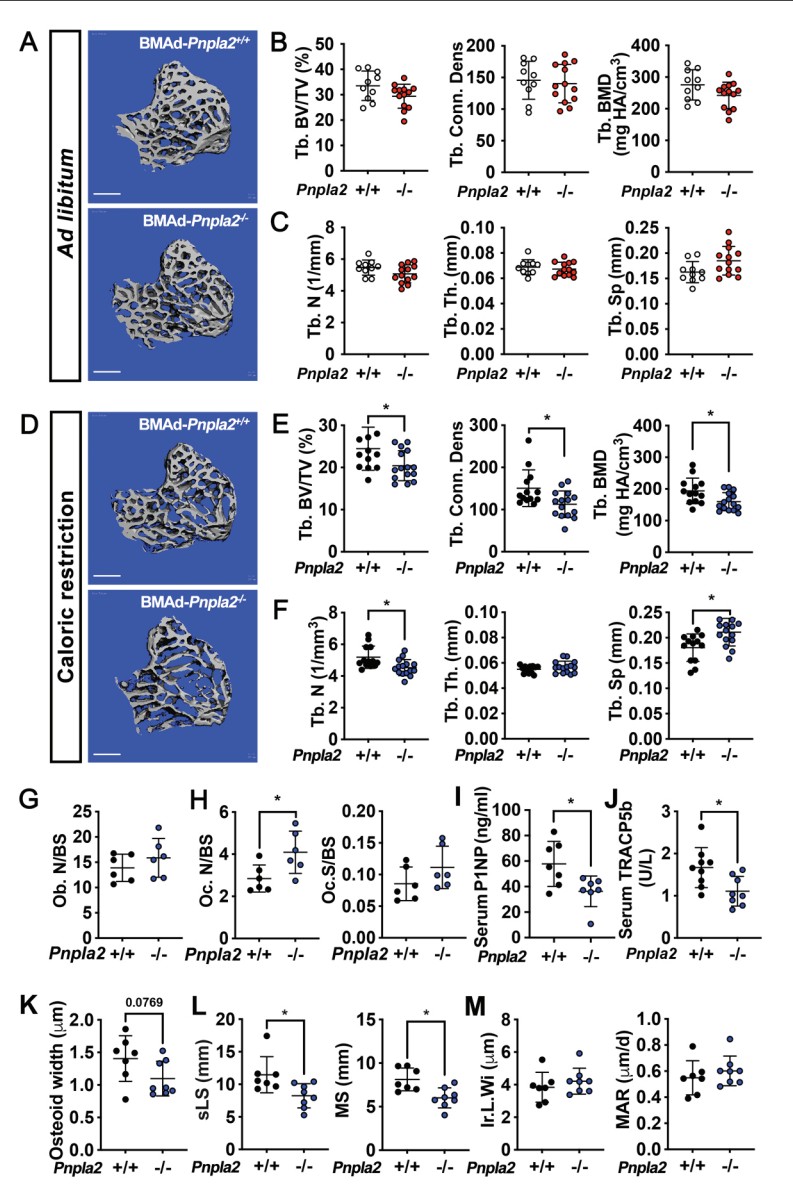

**Figure 3.** BMAd lipolysis is required to maintain bone homeostasis in male mice under CR conditions, but not when mice are fed ad libitum. (**A-C**) Male BMAd-*Pnpla2*$^{-/-}$ and their BMAd-*Pnpla2*$^{+/+}$ littermates with ad libitum feeding were euthanized at 24 weeks of age. Two independent age- and sex- matched cohorts were plotted together. Tibiae from ad libitum mice were analyzed by μCT for indicated trabecular (Tb.) bone variables. BV/TV: bone volume fraction; Conn. Dens: connective density; BMD: bone mineral density; N: number; Th: thickness; Sp: separation. Scale bars indicate 500 μm. (**D-M**) Male mice at 18 weeks of age underwent 30% CR for 6 weeks. Two independent age- and sex- matched cohorts were plotted together for μCT parameters (**D-F**), one of those two cohorts was used for ELISA, and static or dynamic histomorphometry (**G-M**). (**D-F**) Tibiae from CR mice were analyzed by μCT for indicated trabecular bone variables. Scale bar; 500 μm. (**G-H**) Static histomorphometry analyses were performed to calculate osteoblast number (Ob. N), osteoclast number (OC. N) and osteoclast surface (Oc. S) per bone surface (BS). (**I-J**) Concentrations of circulating P1NP and TRACP5b in CR mice were measured. (**K**) Osteoid quantification was performed on undecalcified plastic sections with Goldner's Trichrome staining. (**L-M**) Dynamic histomorphometry was performed on calcein-labelled trabecular bone from proximal tibia. sLS: single-labelled surface; MS: mineralized surface; Ir.L.Wi: inter-label width; MAR: mineral apposition rate. Data are expressed as mean ± SD. * indicates p<0.05 with a two-sample *t*-test. In addition, multiple unpaired t tests had been performed crossing all parameters, p values were adjusted for multiple comparisons using Two-stage step-up (Benjamini, Krieger, and Yekutieli) with FDR method. Adjusted p values are shown in *Figure 3—source data 1*.

*Figure 3 continued on next page*

*Figure 3 continued*

The online version of this article includes the following source data and figure supplement(s) for figure 3:

**Source data 1.** BMAd lipolysis is required to maintain bone homeostasis in male mice under CR conditions, but not when mice are fed ad libitum.

**Figure supplement 1.** Blocking BMAd-lipolysis does not influence global metabolism when mice are fed ad libitum or calorically restricted.

**Figure supplement 1—source data 1.** Blocking BMAd-lipolysis does not influence global metabolism when mice are fed ad libitum or calorically restricted.

**Figure supplement 2.** Cortical bone variables in BMAd-*Pnpla2*-/- mice and other possible mechanisms for bone loss in BMAd-*Pnpla2*-/- CR mice.

**Figure supplement 2—source data 1.** Cortical bone variables in BMAd-*Pnpla2*-/- mice and other possible mechanisms for bone loss in BMAd-*Pnpla2*-/- CR mice.

**Figure supplement 3.** BMAd lipolysis is not required in female mice to maintain bone homeostasis under CR conditions.

**Figure supplement 3—source data 1.** BMAd lipolysis is not required in female mice to maintain bone homeostasis under CR conditions.

**Figure supplement 4.** BMAd-lipolysis impairment in estrogen-deficient female mice does not affect CR-induced bone changes.

**Figure supplement 4—source data 1.** BMAd-lipolysis impairment in estrogen-deficient female mice does not affect CR-induced bone changes.

surface (Oc.S/BS) was not changed and the osteoclast activation marker, TRACP5b, was decreased (*Figure 3H and J*), suggesting osteoclast function was inhibited. No differences were observed in RANK Ligand and CTX-1 with CR (*Figure 3—figure supplement 2K and L*). We then visualized and quantified osteoid thickness with Goldner's trichrome staining and observed a trend towards thinner osteoid layers without affecting osteoid surface (*Figure 3K* and *Figure 3—figure supplement 2M*), which could be secondary to impaired secretion of collagen matrix by osteoblasts or to enhanced bone mineralization. Further, we injected mice with calcein to label those bone surfaces undergoing active mineralization. Dynamic histomorphometry data suggests that BMAd-*Pnpla2*-/- mice had less bone-forming surface, as indicated by reduced single-labelled surface (sLS) and mineral surface (MS) (*Figure 3L*). There were no differences in double-labelled bone surface, inter-label width, mineral apposition rate or osteoid maturation time (*Figure 3M* and *Figure 3—figure supplement 2N*), suggesting that bone mineralization is not affected by loss of BMAd lipolysis. Taken together, these data support a model in which reduced trabecular bone in BMAd-*Pnpla2*-/- mice is likely due to impaired ability of osteoblasts to secrete osteoid.

## BMAd lipolysis is not required to maintain bone homeostasis under calorie-restricted conditions in female mice

To determine whether sex influences responses of BMAd-*Pnpla2*-/- mice to CR, we performed similar experiments in female mice at 20 weeks of age. As expected, after six weeks of CR, both control and knockout mice exhibited comparable reduced body weights, random blood glucose concentrations, and tissue weights (*Figure 3—figure supplement 3A and B*). CR or *Pnpla2* deficiency caused expansion of proximal tibial rBMAT, which was further increased in CR BMAd-*Pnpla2*-/- mice (*Figure 3—figure supplement 3C*). Whereas CR increased trabecular bone BV/TV, and decreased Tb. Sp, BMAd *Pnpla2* deletion did not cause changes in trabecular or cortical bone variables in female mice (*Figure 3—figure supplement 3D and E*). In addition, complete blood cell analyses did not reveal substantial differences in white or red blood cell populations influenced by diet or genotype (*Figure 3—figure supplement 3F*).

We next considered whether estrogen protects female mice from CR-induced osteoporosis. Thus, we ovariectomized control and BMAd-*Pnpla2*-/- mice 2 weeks prior to initiation of CR. Following initiation of CR, both control and knockout mice demonstrated rapid reduction in body weight for 2 weeks, then gradually stabilized during the following ten weeks (*Figure 3—figure supplement 4A*). We did not observe significant differences in glucose tolerance or bone length with CR or *Pnpla2* deletion (*Figure 3—figure supplement 4B and C*). Although CR caused reduction in WAT and spleen weights,

these variables were generally not different between genotypes (*Figure 3—figure supplement 4D*). Interestingly, despite the dramatic increases of BMAT with CR or BMAd-*Pnpla2* deficiency, OVX-associated trabecular bone loss was partially eliminated by CR treatment, which was evidenced by a relative increase in bone volume fraction (Tb. BV/TV) and trabecular number (Tb. N) (*Figure 3—figure supplement 4E and F*). These data suggest that the metabolic benefits of CR diet may combat detrimental effects of estrogen deficiency and/or aging. Although not observed in intact mice, with OVX *Pnpla2*-deficient mice, the loss of BMAd ATGL caused a slight reduction in trabecular bone mineral density (Tb. BMD) and thickness (Tb. Th) compared to OVX controls (revealed by two-way ANOVA analysis) (*Figure 3—figure supplement 4F*).

## BMAd lipolytic deficiency impairs myelopoiesis during regeneration

A single rBMAd connects to almost 100 hematopoietic cells (*Robles et al., 2019*), suggesting that BMAds potentially serve as important local energy sources for hematopoiesis. However, when fed ad libitum, male BMAd-*Pnpla2*$^{-/-}$ mice did not exhibit differences in mature blood cells populations, as assessed by complete blood cell counts (*Supplementary file 1*) or in bone marrow hematopoietic cell populations, as assessed by flow cytometry (*Supplementary file 2* and *Figure 4—figure supplement 1A and B*). However, we did observe a reduced percentage of bone marrow mature neutrophils in CR BMAd-*Pnpla2*$^{-/-}$ mice (*Supplementary file 1*, *Supplementary file 2*), perhaps due to impaired maturation of granulocytes. These data suggest that although neutrophil defects were observed in CR BMAd-*Pnpla2*$^{-/-}$ mice, other hematopoietic cell populations can metabolically compensate for the lack of BMAd lipolytic products.

To investigate further whether hematopoiesis is dependent upon BMAd lipolysis, we administered a sublethal dose of whole-body irradiation, and evaluated hematopoietic cell recovery. Following irradiation, white and red blood cell depletion and recovery were monitored by complete blood cell counts every 2–3 days (*Figure 4—figure supplement 1C-E*). During the recovery phase, three-way ANOVA analyses revealed that differences in time (days) and diet (AL vs CR) were observed for total white blood cells, lymphocytes, and red blood cells, while significant interactions between genotype (BMAd-*Pnpla2*$^{+/+}$ vs BMAd-*Pnpla2*$^{-/-}$) and time were observed for neutrophils, eosinophils and basophils, suggesting BMAd-*Pnpla2* deficiency specifically influenced myeloid lineage cells. Nine days after irradiation, HSPCs (*Figure 4—figure supplement 1F and G*) and mature/immature hematopoietic cells (*Figure 4A–H*) were not influenced by deficiency of BMAd-lipolysis in ad libitum fed mice. However, CR decreased bone marrow cellularity (BMNC) (*Figure 4A*), and most HSPC populations in control mice (*Figure 4—figure supplement 1F and G*). Two-way ANOVA also revealed inhibitory effects of CR on monocytes and neutrophils, without changes to lymphocytes (*Figure 4B–E*). In BMAd-*Pnpla2*$^{-/-}$ mice, HSPCs were not altered by CR except for PreMegE cells, which were suppressed. Bone marrow cellularity, monocytes, and neutrophils were also decreased in CR BMAd-*Pnpla2*$^{-/-}$ mice. The recovery of neutrophils, preneutrophils, immature and mature neutrophils following irradiation was impaired by CR, and *Pnpla2*-deficiency further reduced ability of neutrophils and mature neutrophils to regenerate (*Figure 4E–H*). These reductions in monocytes and neutrophils in CR BMAd-*Pnpla2*$^{-/-}$ mice suggest that BMAd-lipolysis is required for myeloid cell lineage regeneration when energy supply from circulation is limited. CFU assays, which were optimized for the growth of myeloid progenitor cells (CFU-Granulocytes, CFU-Macrophages and CFU-GM), demonstrated a trend towards fewer granulocyte progenitor colonies (CFU-G) from CR BMAd-*Pnpla2*$^{-/-}$ mice (*Figure 4I*). The proliferative capacity of macrophage progenitors (CFU-M) was impaired when derived from BMAd-*Pnpla2*$^{-/-}$ mice fed ad libitum, or when derived from CR mice of either genotype (*Figure 4J*). CR marrow produced fewer granulocyte-macrophage progenitors (CFU-GM), which was further reduced when isolated from mice with impaired BMAd-lipolysis (*Figure 4K*). These data suggest that proliferative and differentiation capacities of cultured myeloid cell progenitors may have been reprogrammed in the marrow niche when their energy supply was limited either by CR or impaired BMAd lipolysis (*Figure 4L*).

## Coupling of BMAd *Pnpla2* deletion and CR results in extensive alterations to the bone marrow transcriptome

To determine mechanisms by which BMAd *Pnpla2* deficiency causes bone loss in CR male mice, we profiled overall gene expression using bulk RNA sequencing (RNAseq) in bone marrow plugs from distal tibiae, a skeletal location highly enriched with BMAT. PCA plots showed that RNA profiles from

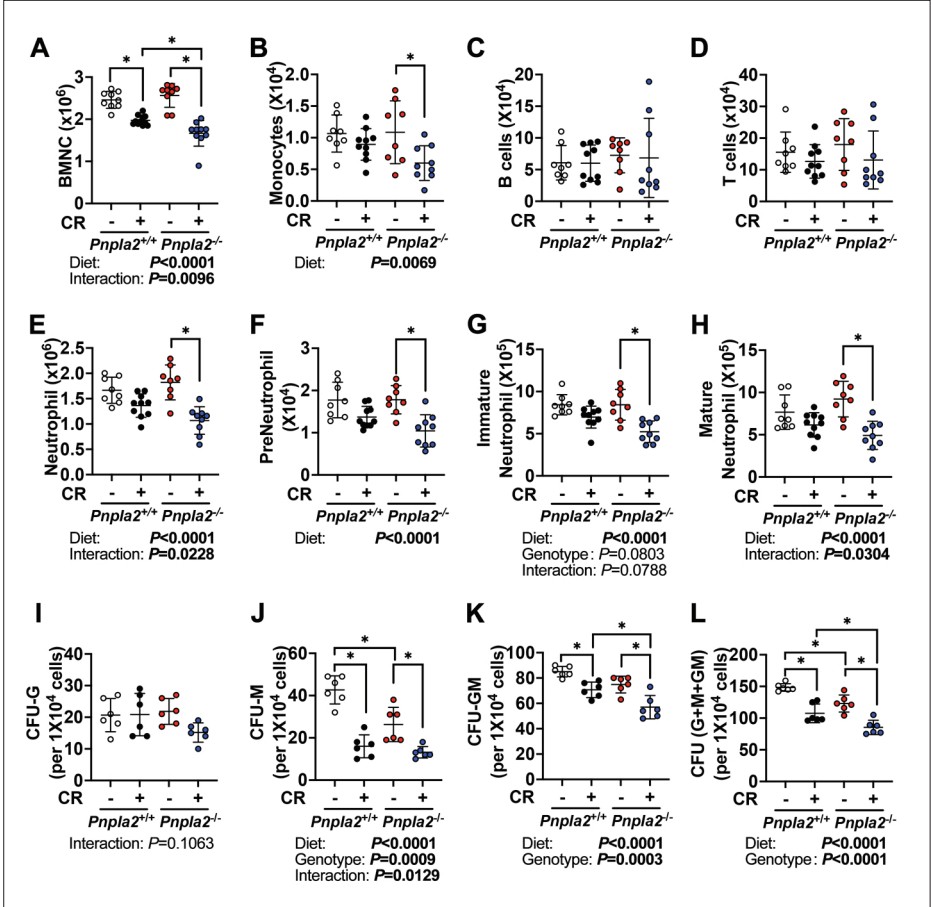

**Figure 4.** BMAd-*Pnpla2* deficiency impairs myelopoiesis. (**A-H**) BMAd-*Pnpla2*⁻/⁻ mice and littermate controls (*Pnpla2*⁺/⁺) were caloric restricted (CR; +) for 20 weeks or remained on an ad libitum diet (-), and then received whole-body irradiation (6 Gy). Mice were euthanized 9 days post-irradiation. Femurs were collected for flow cytometry to measure the regeneration of hematopoietic cells. Bone marrow mononuclear cells (BMNCs), monocytes, B and T lymphocytes and neutrophils were quantified. (**I-L**) CFU assays. Femora and tibial bone marrow cells were isolated from BMAd-*Pnpla2*⁻/⁻ mice and littermate controls (*Pnpla2*⁺/⁺), which had been fed ad libitum (-) or a caloric restricted (CR; +) diet for 20 weeks. After counting, 1 × 10⁴ cells were plated for CFU assays. Colonies were counted by an independent expert in a blinded manner 7 days after plating. Data are expressed as mean ± SD. * indicates p<0.05 with two-way ANOVA analyses followed by Šídák's multiple comparisons test. Significant effects of genotype, diet, or their interactions are shown, as are trends.

The online version of this article includes the following source data and figure supplement(s) for figure 4:

**Source data 1.** BMAd-*Pnpla2* deficiency impairs myelopoiesis.

**Figure supplement 1.** Flow cytometry strategies for hematopoietic cells and sublethal irradiation-induced hematopoietic regeneration in BMAd-*Pnpla2*⁻/⁻ mice.

**Figure supplement 1—source data 1.** Flow cytometry strategies for hematopoietic cells and sublethal irradiation-induced hematopoietic regeneration in BMAd-*Pnpla2*⁻/⁻ mice.

CR groups were distinct from ad libitum controls (*Figure 5—figure supplement 1A*). Whereas *Pnpla2* deficiency did not cause gene expression to diverge substantially in mice fed ad libitum, loss of *Pnpla2* interacted with CR to cause a well-segregated pattern of gene expression (*Figure 5—figure supplement 1A*). By our criteria (padj <0.05, |log2 fold change|>1), CR changed expression of 1026 genes compared to ad libitum controls (left). Although *Pnpla2* deficiency alone only altered 10 genes in BMAd-*Pnpla2*⁻/⁻ mice fed ad libitum (middle), loss of *Pnpla2* in CR mice caused alterations in 1060 genes (right) (*Figure 5—figure supplement 1B*). Analyses of genes regulated in BMAd-*Pnpla2*⁺/⁺ mice with CR revealed four distinct clusters (*Figure 5A*). Approximately 80% of genes fell in cluster 1, which contained genes upregulated by CR in control mice, with induction largely blocked by *Pnpla2*

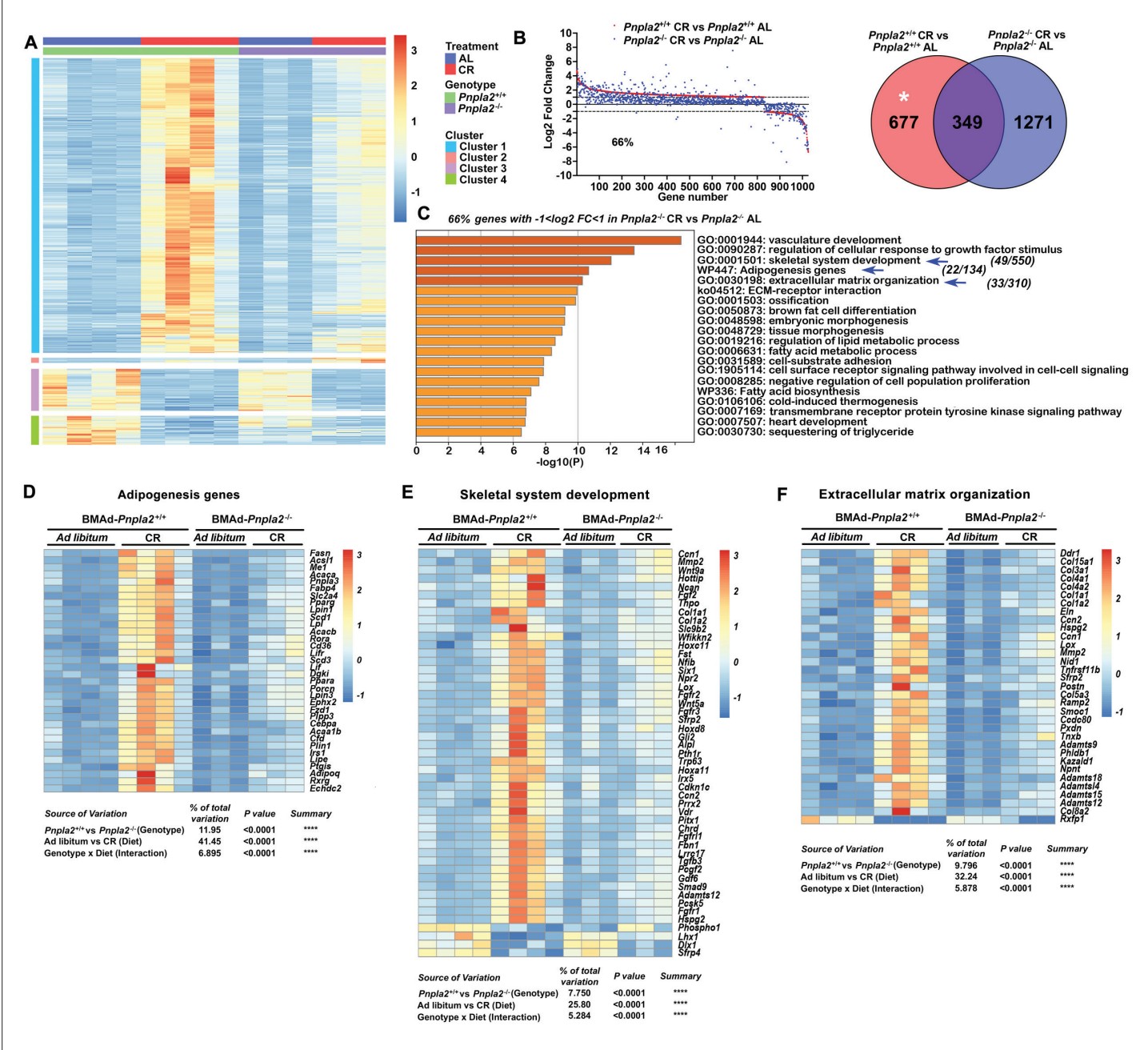

**Figure 5.** BMAd-*Pnpla2* deficiency causes extensive alterations to the bone marrow transcriptome only when coupled with CR. Male control and BMAd-*Pnpla2*$^{-/-}$ mice at 24 weeks of age were either fed AL or underwent 30% CR for 6 weeks. Distal tibial cBMAT was flushed and cBMAT from two mice was pooled as one sample for RNAseq analyses (n of 3 or 4 per treatment). (**A**) Differential genes with our criteria (padjj <0.05 and |Log2 fold change|>1) between BMAd-*Pnpla2*$^{+/+}$ CR and BMAd-*Pnpla2*$^{+/+}$ ad libitum (AL) were grouped into 4 clusters. (**B**) Genes different between BMAd-*Pnpla2*$^{+/+}$ CR and BMAd-*Pnpla2*$^{+/+}$ AL were ordered from maximum to minimum log2 fold change (red dots), and compared to corresponding data from BMAd-*Pnpla2*$^{-/-}$ CR versus BMAd-*Pnpla2*$^{-/-}$ AL (blue dots). Venn diagram shows the differential genes between BMAd-*Pnpla2*$^{+/+}$ CR versus BMAd-*Pnpla2*$^{+/+}$ AL BMAT; and BMAd-*Pnpla2*$^{-/-}$ CR versus BMAd-*Pnpla2*$^{-/-}$ AL BMAT. (**C**) Pathway analyses of genes significantly changed by CR in BMAd-*Pnpla2*$^{+/+}$ mice, but not in CR mice lacking *Pnpla2* (indicated by * area in panel **B**). Pathways further analyzed by heatmap indicated with blue arrows. (**D-F**) Expression Z-scores of genes related to adipogenesis (**D**), skeletal system development (**E**) and extracellular matrix organization (**F**) were shown as heatmap. Effects of genotype and diet, and their interactions were analyzed by three-way ANOVA.

The online version of this article includes the following source data and figure supplement(s) for figure 5:

**Source data 1.** BMAd-Pnpla2 deficiency causes extensive alterations to the bone marrow transcriptome only when coupled with CR.

**Figure supplement 1.** CR causes profound changes in BMAT transcriptome.

*Figure 5 continued on next page*

*Figure 5 continued*

**Figure supplement 2.** BMAd-*Pnpla2* deficiency causes extensive alterations to the bone marrow transcriptome only when coupled with CR.

**Figure supplement 2—source data 1.** BMAd-*Pnpla2* deficiency causes extensive alterations to the bone marrow transcriptome only when coupled with CR.

**Figure supplement 3.** BMAd-*Pnpla2* deficiency alters gene expression in response to CR.

deficiency in CR mice. Pathway analyses of cluster 1 with Metascape (https://metascape.org) revealed that regulated genes were associated with vasculature development and cellular response to growth factor stimulus, which likely reflects adaptation mechanisms to compensate for energy insufficiency (*Figure 5—figure supplement 1C*). In addition, genes associated with skeletal system development, extracellular matrix organization, and adipogenesis pathways were upregulated by CR in control mice, but these effects were blunted by *Pnpla2* deletion (*Figure 5—figure supplement 1C*). Cluster 2 includes genes that were mildly upregulated by CR in control mice and were further increased with *Pnpla2*-deficiency. Although no specific pathways were enriched in this gene set, upregulation of two monocarboxylate transporters, *Slc16a11* and *Slc16a4*, suggests a potential mechanism for fueling cellular metabolism, and perhaps for the energy source underlying expansion of BMAd with CR and BMAd *Pnpla2* deletion (*Figure 5—figure supplement 1D*). Cluster 3 highlighted genes that were downregulated by CR treatment regardless of genotype (*Figure 5—figure supplement 1E*), and that were associated with B cell proliferation and interleukin-8 production, which may partially explain changes in hematopoietic cellularity. Cluster 4 contained 80 genes that were down-regulated independently by CR and *Pnpla2*-deficiency, and were associated with regulation of ossification and calcium-mediated signaling (*Figure 5—figure supplement 1F*).

To evaluate further whether BMAd-*Pnpla2* is required for the adaptation of BMAT to CR, we graphically ordered genes from those maximally induced to those most repressed by CR in control mice (*Figure 5B*; red dots). Of these genes, 66% did not meet our criteria for regulated expression by CR in BMAd-*Pnpla2*$^{-/-}$ mice (blue dots). Pathway analyses on the 34% of genes regulated by CR regardless of *Pnpla2* deficiency revealed association with ribonuclease activity, response to hormones and other transmembrane signaling pathways (*Figure 5—figure supplement 2A*). Pathway analyses of genes for which *Pnpla2* is required for response to CR identified vasculature development and cellular response to growth factor stimulus, similar to cluster 1, followed closely by skeletal system development, adipogenesis genes, and extracellular matrix organization pathways (*Figure 5C*). A heatmap showed that the adipogenic genes, including *Adipoq, Fabp4, Cebpa, Lipe, Lpl, Ppara, Pparg, Scd1, Plin1, Cd36, Fasn,* and *Acaca,* were upregulated by CR in control mice (*Figure 5D*), a subset of which were confirmed by qPCR (*Figure 5—figure supplement 2B*). These changes may help to explain molecular mechanisms underlying BMAT expansion following CR. It is important to note that whereas CR BMAd-*Pnpla2*$^{+/+}$ and BMAd-*Pnpla2*$^{-/-}$ mice had comparable amounts of BMAT (*Figure 3—figure supplement 1O and P*), adipocyte gene expression was greatly suppressed in mice with BMAT lacking *Pnpla2* (*Figure 5D*). This observation is consistent with prior work showing that adipocyte-specific deletion of *Pnpla2* results in decreased expression of genes associated with lipid uptake, synthesis, and adipogenesis (*Schoiswohl et al., 2015*), perhaps because NEFA and associated metabolites act as PPAR ligands (*Mottillo et al., 2012*). In addition, genes related to endogenous fatty acid biosynthesis were also increased by CR in BMAd-*Pnpla2*$^{+/+}$ mice, but not in mice lacking *Pnpla2* (*Figure 5—figure supplement 2C*). Interestingly, myeloid leukocyte differentiation genes followed a similar pattern (*Figure 5—figure supplement 2D*), which may contribute to the reduced neutrophil production and impaired myeloid cell proliferation of CR BMAd-*Pnpla2*$^{-/-}$ mice (*Supplementary file 1, Supplementary file 2,* and *Figure 4*).

We previously observed increased bone loss in BMAd-*Pnpla2*$^{-/-}$ mice challenged with CR (*Figure 3*). To investigate potential mechanisms underlying this bone loss, we analyzed pathways from Cluster 1 related to bone metabolism, skeletal system development and extracellular matrix organization. Interestingly, osteoblast-derived alkaline phosphatase (*Alpl*), *Col1a1* and *Col1a2,* and bone marrow Fgf/ Fgfr and Wnt signaling-related molecules were highly induced by CR in control mice but not in *Pnpla2*-deficient mice (*Figure 5E*); many of these genes are also found in the ossification pathway. Multiple collagen genes, *Lox,* and *Adamts* (A Disintegrin and Metalloproteinase with Thrombospondin motifs) family members, which are multidomain extracellular protease enzymes and play key roles in extracellular matrix remodeling, were also upregulated by CR in control mice, with effects largely eliminated

by *Pnpla2* deficiency (**Figure 5F**). These findings may partially explain why osteoid thickness tended to be thinner in BMAd-*Pnpla2*⁻ᐟ⁻ mice (**Figure 3K**). In this regard, 20 collagen genes were significantly up-regulated in control mice following CR (**Figure 5—figure supplement 2E**), whereas only four collagen genes were elevated in CR BMAd-*Pnpla2*⁻ᐟ⁻ mice. Taken together, these data suggest that under conditions of limited dietary energy, BMAds provide energy to maintain osteoblast functions, including the secretion of collagen matrix for osteoid synthesis.

As mentioned above, loss of *Pnpla2* in CR mice altered expression of 1060 genes (**Figure 5—figure supplement 1B**; right), which were clustered into three groups (**Figure 5—figure supplement 3A**). Cluster 1 included genes that were induced only in CR BMAd-*Pnpla2*⁻ᐟ⁻ mice. Although not localized to specific pathways, these genes were involved in diverse metabolic and cellular processes (e.g. *Perm1*, *mt-Nd6*, and *B3gat2*), and were generally expressed at low levels at baseline. In contrast, cluster 2 contained genes that were suppressed in CR BMAd-*Pnpla2*⁻ᐟ⁻ mice, including pathways related to mitochondrial functions, hematopoietic progenitor cell differentiation, positive regulation of osteoblast differentiation and fatty acid biosynthetic process (**Figure 5—figure supplement 3B and C**), which sheds further light on the impaired HSPC proliferative capacity and osteoblast functions. Genes in cluster 3 were upregulated by CR, but not by CR in the absence of BMAd-*Pnpla2*. Pathway analyses highlighted the regulation of lipid catabolic process, triacylglyceride synthesis and leukocyte migration (**Figure 5—figure supplement 3D and E**), consistent with the impaired lipid metabolism and leukocyte production observed in BMAd-*Pnpla2*⁻ᐟ⁻ mice.

## BMAd lipolysis is required for trabecular and cortical bone regeneration

To test whether BMAds are a critical source of local energy under conditions where energy needs are elevated, we investigated the impact of CR and BMAd *Pnpla2* deficiency on bone regeneration. To do this, we created a 0.7 mm hole in the proximal tibia, approximately 1–2 mm distal from the growth plate, and evaluated the newly formed woven bone in both the trabecular (endocortical) and cortical compartments nine days later (**Figure 6A**). As expected, CR impaired formation of bone in the trabecular (endocortical) region of interest (ROI), resulting in decreased bone volume fraction, mineral density, and mineral content (**Figure 6B**). Importantly, these effects of CR were mimicked by BMAd-specific *Pnpla2* deletion; however, gene deletion did not compound effects of CR on impaired bone regeneration, perhaps because newly-formed bone volume was already low with either CR or *Pnpla2* deficiency alone. We then visualized newly formed bone in the cortical bone compartment with Safranin O/ Fast Green (SO/FG) staining of paraffin-embedded proximal tibia sections (**Figure 6C**). Of note, both bone marrow stromal and periosteal cells contribute to the cortical bone regeneration and are derived from common mesenchymal progenitors (**Duchamp de Lageneste et al., 2018**). Although deficiency of BMAd-lipolysis is unlikely to affect periosteal cell functions when the cortical bone is intact, it may interact with this cell population during bone regeneration. CR also impaired formation of new woven bone in the cortical compartment by decreasing bone volume fraction, mineral density, and mineral content. As with bone formation in the trabecular compartment, effects of CR on the cortical compartment are mimicked by BMAd-specific depletion of *Pnpla2*, and effects of impaired BMAd lipolysis did not exacerbate CR-induced impaired bone regeneration (**Figure 6D**). Taken together, these data provide compelling evidence that under conditions where energy requirements are high, BMAd provide a critical local energy source for early-stage bone regeneration.

## Energy from BMAd protects against bone loss caused by chronic cold exposure

Cold exposure is well-documented to increase energy expenditure and adaptive thermogenesis, largely fueled by energy stored in WAT depots. To explore further under what conditions BMAd lipolysis may be critical for bone cell functions, we first evaluated female control and BMAd-*Pnpla2*⁻ᐟ⁻ mice, which have higher content of BMAT than males, and housed them at room (22 °C) or cold (4 °C) temperatures for three weeks when they were at 20 weeks of age. As expected (**Scheller et al., 2015**; **Scheller et al., 2019**), cold exposure resulted in smaller rBMAds within the proximal tibia of BMAd-*Pnpla2*⁺ᐟ⁺ mice (**Figure 6E and F**). In contrast, rBMAd size was increased at baseline in BMAd-*Pnpla2*⁻ᐟ⁻ mice, with no reduction observed with cold exposure (**Figure 6E and G**). These data indicate that intact BMAd lipolysis was required for reduction of BMAd size with cold exposure. We then

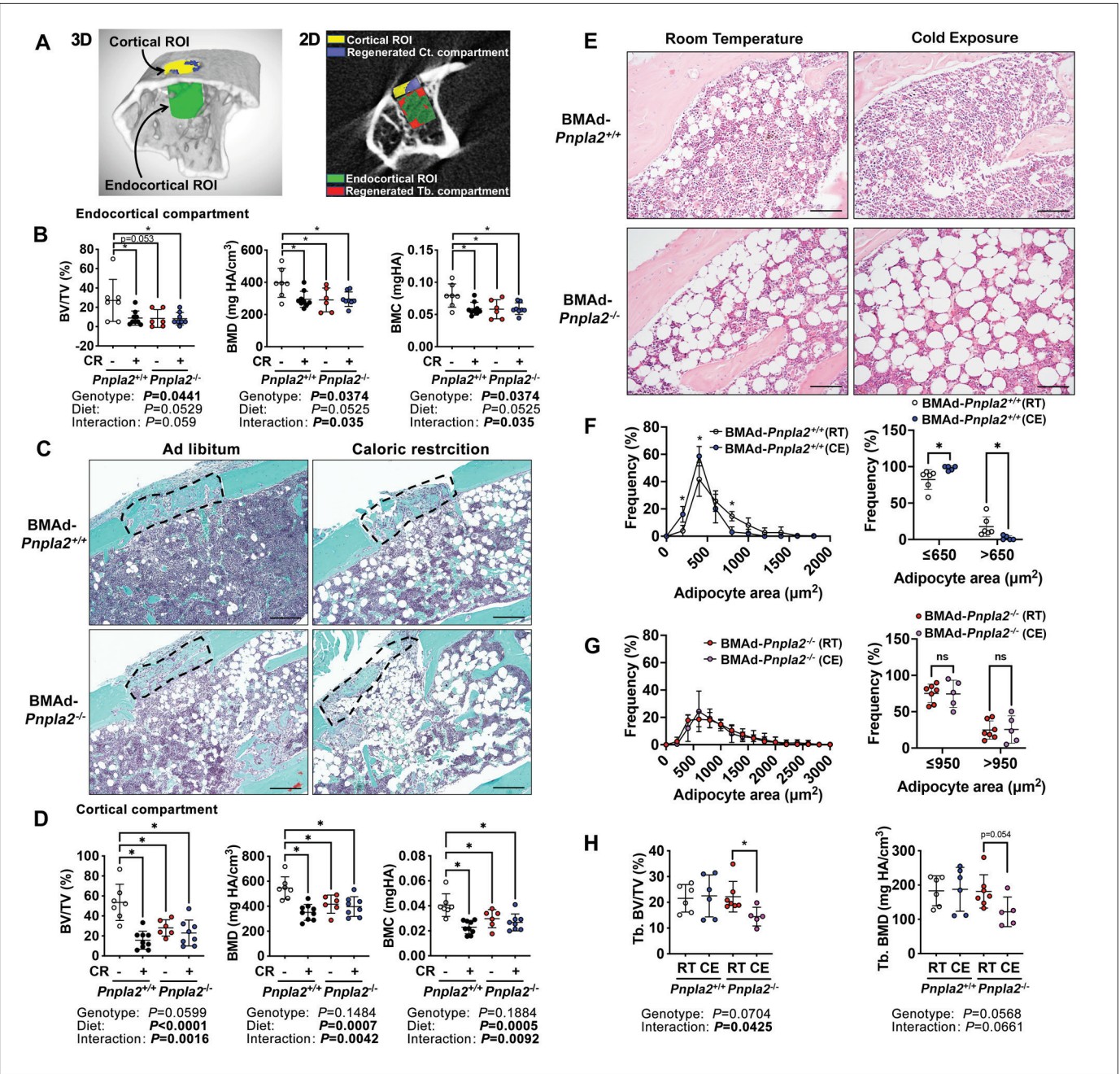

**Figure 6.** Energy from BMAd is required for trabecular bone regeneration and protects against bone loss caused by chronic cold exposure. (**A-D**) BMAd-*Pnpla2+/+* and BMAd-*Pnpla2-/-* male mice at 24 weeks of age fed with chow diet (-) or underwent 30% CR for six weeks (+). A 0.7 mm proximal tibial defect was created 1–2 mm distal to the growth plate. Tibiae were collected 9 days after surgery. MicroCT was performed to analyze trabecular (endocortical) and cortical bone regeneration. (**A**) Representative analyzing images of bone defect. Example of μCT showing trabecular (endocortical) region of interest (ROI; green) and cortical ROI (yellow). Newly generated woven bone in trabecular (endocortical; red) and cortical compartment (purple) is indicated. (**B**) Quantification of regenerated bone in the endocortical compartment - bone volume fraction (BV/TV), mineral density (BMD) and mineral content (BMC). Data are expressed as mean ± SD. (**C**) Safranin O/ Fast Green (SO/FG) staining of new cortical compartment formation in defect sites. Scale bar; 200 μm. (**D**) Quantification of regenerated cortical woven bone volume fraction (BV/TV), mineral density (BMD) and mineral content (BMC). (**E-I**) Female BMAd-*Pnpla2+/+* and BMAd-*Pnpla2-/-* mice at 20 weeks of age were singly housed at 4°C for three weeks without enrichments. Tibiae were collected for sectioning and μCT analyses. (**E**) Proximal tibiae were decalcified and paraffin-sectioned for H&E staining. Representative images for proximal tibia are shown. Scale bar, 100 μm. (**F-G**) Quantification of BMAds from H&E-stained slides using MetaMorph software. Comparison of BMAd size at room temperature (RT) versus cold exposure (CE) in mice of indicated genotypes. (**H**) Trabecular bone volume fraction (BV/TV) and mineral density (BMD) were quantified by μCT. Data are expressed as mean ± SD. * indicates p<0.05 with two-way ANOVA analyses followed by Šídák's multiple comparisons test. Significant effects of genotype, diet, or their interactions are shown, as are trends.

*Figure 6 continued on next page*

*Figure 6 continued*

The online version of this article includes the following source data and figure supplement(s) for figure 6:

**Source data 1.** Energy from BMAd is required for trabecular bone regeneration and protects against bone loss caused by chronic cold exposure.

**Figure supplement 1.** Energy from BMAd protects against changes in trabecular microstructures caused by chronic cold exposure.

**Figure supplement 1—source data 1.** Energy from BMAd protects against changes in trabecular microstructures caused by chronic cold exposure.

evaluated bone mass by μCT and found that whereas trabecular bone of the proximal tibia was maintained in BMAd-*Pnpla2*^+/+ mice with cold stress, trabecular bone volume fraction declined with cold exposure in BMAd-*Pnpla2*^-/- mice and bone mineral density showed a similar trend (*Figure 6H*). These results suggest that lipolysis from vicinal BMAds is important for maintenance of bone when energy needs are high or when energy supply is limited. Histomorphometry analyses did not reveal significant changes in osteoblast and osteoclast numbers, or in TRAP-positive osteoclast surface per bone surface (*Figure 6—figure supplement 1A-C*). However, the bone formation marker, P1NP, decreased in cold-exposed BMAd-*Pnpla2*^-/- mice compared to room temperature controls (*Figure 6—figure supplement 1D*), suggesting that osteoblast functions were impaired.

We also evaluated whether BMAT lipolysis is necessary for maintenance of trabecular bone microstructures in male mice under cold exposure conditions (*Figure 6—figure supplement 1E-J*). Interestingly, control BMAd-*Pnpla2*^+/+ male mice were more vulnerable to cold-induced bone loss than female mice, with significant reductions of trabecular bone volume fraction, mineral density, and thickness. There was no obvious further decrease of these parameters in BMAd-*Pnpla2*^-/- mice, perhaps because bone volume was already low with cold exposure. However, two-way ANOVA analyses revealed there were interactions between temperature and genotype, suggesting there were further changes in BMAd-*Pnpla2*^-/- mice (*Figure 6—figure supplement 1E,F,H* ). Although cold stimuli did not affect the microstructure parameters of trabecular bone in BMAd-*Pnpla2*^+/+ mice, cold exposure decreased trabecular connective density and trabecular number, and increased trabecular bone spacing in BMAd-*Pnpla2*^-/- mice (*Figure 6—figure supplement 1G,I,J* and ). Taken together, these data highlight sexual dimorphism in the response of bone to cold, and that in both males and females, a further loss of bone is observed when BMAT lipolysis is impaired.

## Discussion

In the current study, we developed and tested for the first time, a BMAd-specific mouse model to more clearly understand the function of bone marrow adipose tissue, particularly during times of nutritional, hormonal, mechanical and environmental stress. Previously, the role of BMAds in the marrow niche was deduced through indirect inferences from associations of skeletal and hematopoietic phenotypes with bone marrow adipose tissue volume. This has been exemplified by a number of BMAT depletion models which showed high bone mass, including A-ZIP (*Naveiras et al., 2009*), *Adipoq*-driven DTA (*Zou et al., 2019*), and *Adipoq*-driven loss of *Pparg* (*Wang et al., 2013*), *Lmna* (*Corsa et al., 2021*), or *Bscl2* (*Mcilroy et al., 2018*); however these lipodystrophic mice also lacked white and brown adipose depots and thus exhibit global metabolic dysfunction, including fatty liver, hyperlipidemia and insulin resistance. Moreover, *Adipoq*-driven Cre used in those studies also causes recombination in bone marrow stromal cells (*Mukohira et al., 2019*). Similarly, mouse models or treatments that result in BMAT expansion, such as *Prx1*-driven *Pth1r* knockout mice (*Fan et al., 2017*), CR, thiazolidinedione administration, estrogen-deficiency, and irradiation (*Li and MacDougald, 2021*), also caused bone loss, but again, effects on bone may be secondary to lack of promoter specificity or confounding systemic effects. Lineage tracing studies have previously been performed to determine cell-specific markers for BMAds. For example, both *Prx1* and *Osterix* are restricted to bone and trace to 100% of BMAds, but are also expressed in mesenchymal cells (*Logan et al., 2002*; *Mizoguchi et al., 2014*). *Nestin* and leptin receptor (*Lepr*) label over 90% of BMAds, but also trace to stromal cells (*Zhou et al., 2017*). Whereas Pdgf-receptor α (*Pdgfrα*)-driven Cre expression causes recombination in all white adipocytes, only 50–70% of BMAds are traced (*Horowitz et al., 2017*). Thus, our successful strategy for targeting BMAds using dual expression of *Osterix* and *Adipoq*, as reported in this work, improves the interpretability of experiments on the distinct roles of BMAds in the marrow niche and will be a useful tool for future investigations.

After generating this novel BMAd-specific Cre mouse model, we successfully ablated *Pnpla2* in BMAds, and demonstrated the necessity of ATGL in BMAd lipolysis. Our studies directly demonstrate for the first time the importance of BMAd lipolysis in myelopoiesis and bone homeostasis under conditions of energetic stress, including CR, irradiation, bone regeneration and cold exposure. With the loss of peripheral WAT in CR mice, there is a dramatic increase in BMAT throughout the tibia, and this is accompanied by increased expression of adipocyte genes, including those involved in lipid uptake, de novo lipogenesis, and lipolysis. BMAT expansion with CR has been observed in both rodents and humans (*Cawthorn et al., 2014*; *Devlin et al., 2010*; *Fazeli et al., 2021*), and mechanisms underlying this observation are still not fully understood. We speculate that CR promotes lipolysis in peripheral adipose tissues, and the released non-esterified fatty acids have increased flux to bone marrow, where they are used either directly to maintain hematopoiesis and bone homeostasis, or used indirectly after having been stored and released from BMAT. The RNAseq data also suggests that BMAds have elevated rates of lipid uptake, lipogenesis and lipolysis with CR in control mice. However, when BMAd lipolysis is impaired, this dynamic cycle is halted, which is reflected by the blunted induction of adipocyte genes in response to CR. This BMAd quiescence causes a shortfall in local energy supply and contributes to hematopoietic cell and osteoblast dysfunction. In this regard, two major collagens secreted by osteoblasts, *Col1a1* and *Col1a2*, and *Alpl* were increased by CR in control BMAd-*Pnpla2*+/+ mice, but not in calorie-restricted BMAd-*Pnpla2*-/- mice. These data are consistent with the trend for thinner osteoid observed in CR BMAd-*Pnpla2*-/- mice and reduced circulating concentrations of the bone formation marker P1NP. Although bone mineralization is not impaired, new bone forming surface is reduced in CR mice that lack BMAd lipolysis. In addition, the number of osteoclasts on trabecular bone surface is higher in CR BMAd-*Pnpla2*-/- mice; however, osteoclast surface per bone surface and a circulating marker of bone turnover (CTX-1) are not increased. A serum marker for osteoclast activity, TRACP5b, is decreased in BMAd-*Pnpla2*-/- mice either fed ad libitum or a CR diet. Taken together, our studies suggest that bone mass reduction in CR BMAd-*Pnpla2*-/- mice is less likely due to degradation of bone by osteoclasts, and more likely due to impaired osteoblast function, including collagen secretion and osteoid production.

Interestingly, the contributions of BMAd lipolysis to bone homeostasis appear to be more important in male mice compared to females. Although we considered that a stronger phenotype might be revealed in female mice following estrogen depletion, the low bone mass observed with ovariectomy or CR may represent a critical threshold that is strongly defended through mechanisms independent of BMAd lipolysis. Alternatively, it is possible that androgens increase energy requirements of bone such that male mice are more dependent on BMAd lipolysis under stressful conditions. Consistent with this tenet, in both mouse and human studies of calorie restriction, males tend to have greater induction of BMAds than females (*Fazeli et al., 2021*). Importantly, we observed that bone volume fraction and trabecular number were increased by 12 weeks of CR in OVX mice of both genotypes, suggesting that the metabolic and anti-aging benefits of CR somehow block the bone loss associated with estrogen deficiency as mice aged to 40 weeks.

Surprisingly, despite expansion of BMAT in ad libitum-fed BMAd-*Pnpla2*-/- male and female mice, we did not observe differences in skeletal parameters, indicating that expansion of BMAT is not sufficient to cause bone loss, and that the correlation between elevated fracture risk and expansion of BMAT under a variety of clinical situations is not necessarily a causal relationship. The critical role of BMAd lipolysis in fueling osteoblasts was observed in BMAd-*Pnpla2*-/- male mice only when energy needs were high, or when the availability of dietary energy was low. Of note, cellular protein synthesis typically uses 25–30% of the oxygen consumption coupled to ATP synthesis (*Rolfe and Brown, 1997*), and this percentage may be higher in osteoblasts since protein synthesis and translation are integral to osteoblast function. Additionally, fatty acids contribute substantially to the energy demands of bone tissue and cells (*Adamek et al., 1987*), and in the absence of BMAd lipolysis, these energy needs must be met from circulating lipids, glucose, lactate, and amino acids. Uptake of fatty acids by osteoblasts is likely mediated by the CD36 fatty acid translocase, which is required in mice to maintain osteoblast numbers and activity (*Kevorkova et al., 2013*). Further, impairment of fatty acid oxidation in osteoblasts and osteocytes led to reduced postnatal bone acquisition in female, but not male mice. Interestingly, significant increases in the osteoid thickness, osteoid volume per bone volume, and the osteoid maturation time suggest that a mineralization defect occurs in mice unable to oxidize fatty acids obtained not only from BMAds, but also from circulation (*Kim et al., 2017*). Although we

have largely interpreted effects of Pnpla2 deficiency as causing a local energy shortfall, *Pnpla2* deficiency may also alter downstream signaling activities. For example, ATGL deficiency prevents cAMP-dependent degradation of TXNIP and thus reduces glucose uptake and lactate secretion (*Beg et al., 2021*). ATGL-catalyzed lipolysis also provides essential mediators to increase activity of PGC-1α/PPARα and PPARδpathways (*Haemmerle et al., 2011*; *Khan et al., 2015*). Thus, it is conceivable that ATGL deficiency may alter the BMAd transcriptome and secretome to influence the marrow niche independent of its role as a fuel source.

Given the extensive literature describing interactions between BMAds and hematopoietic cells (*Lee et al., 2018*; *Valet et al., 2020*), we were surprised at the lack of substantial changes in hematopoietic progenitors, white blood cells, and red blood cells in BMAd-*Pnpla2⁻/⁻* mice under basal and CR conditions. One possibility is that skeletal sites containing low BMAd numbers, including vertebrae, sternum, ribs and pelvis, may allow compensatory formation of circulating mature blood cells (*Kricun, 1985*). In addition, hematopoietic progenitors and mature blood cell populations were not substantially altered in bone marrow of long bones, with the exception of myeloid differentiation into neutrophils, suggesting that hematopoietic cellularity is generally maintained despite expansion of BMAT volume. Although hematopoietic progenitors and mature cells can use lipids for energy, glucose serves as the major source of energy for glycolysis and oxidative metabolism in these cell populations (*Jeon et al., 2020*; *Roy et al., 2018*). It is perhaps unsurprising, given the importance of blood cell production in maintaining life, that there is a great deal of metabolic flexibility when it comes to fuel sources for hematopoiesis. While HSPC numbers are unaltered in our BMAd-*Pnpla2⁻/⁻* mice, myeloid lineage cell recovery is significantly blunted by CR and deficiency of BMAd-lipolysis following sublethal irradiation and in in vitro myeloid CFU assays. Myeloid progenitors from CR BMAd-*Pnpla2⁻/⁻* mice had an impaired capacity to expand, suggesting that progenitors were reprogrammed in response to energy deficiency in the bone marrow niche. We did not profile the metabolic changes of HSPCs in CR BMAd-*Pnpla2⁻/⁻* mice, but it has been reported that fatty acid oxidation is required for HSC asymmetric division to retain the stem cell properties (*Ito et al., 2012*).

In summary, we have developed a novel mouse model to specifically evaluate the importance of BMAds in adult life and during times of nutritional, hormonal, environmental and mechanical stress. We report that BMAds are a local energy source that support myeloid cell lineage regeneration following irradiation, and maintain progenitor differentiation capacity when systemic energy is limiting. In addition, we find that BMAT serves a highly specialized function to maintain bone mass and osteoblast function in times of elevated local energy needs such as with bone regeneration, increased whole body energy needs from cold exposure, or when dietary energy is limited due to CR. Thus, Steele's instincts in 1884 when he wrote that fat from marrow 'nourishes the skeleton' are largely borne out by modern experimentation.

## Limitations of study

There are some limitations to the BMAd-*Cre* mice that should be noted. For instance, expression of *Cre* from the IRES within the 3'-UTR of *Adipoq* is relatively low, and thus rates of recombination are less frequent in young mice than optimal. Mice that are older than 16 weeks are suggested for future usage of this *Cre* mouse model. However, bone formation and turnover rates decrease with age both in mice and humans (*Fatayerji and Eastell, 1999*; *Ferguson et al., 2003*), which may provide challenges for mechanistic studies. The IRES-*Cre* cassette also causes hypoadiponectinemia so experimental design must include appropriate controls with BMAd-*Cre* positive mice with or without floxed genes. Although a total absence of adiponectin reduces bone mass (*Naot et al., 2016*; *Yang et al., 2019*; *Zhu et al., 2020*), mice with hypoadiponectinemia described here did not exhibit metabolic or bone phenotypes. In retrospect, a better approach might have been to accept loss of one *Adipoq* allele and insert the FAC cassette at the start site of the adiponectin coding region to promote high levels of *Cre* expression. This strategy might also have allowed inducible knockout of BMAd genes with tamoxifen through expression of CreERT2, which we found to be too inefficient to be functional when inserted into the 3'-UTR of *Adipoq*, with only 5–8% BMAds labeled in tamoxifen-treated adult mice. Of note, all mice used in these studies were on a mixed SJL and C57BL/6 J background, and mouse strain influences bone mass, and responsiveness of bone to stressors. Although we have not provided direct evidence of BMAd-derived fatty acids fueling osteoblasts or hematopoietic cells, this line of investigation will require technical advances to distinguish BMAd-derived NEFA from those in

circulation. It should be noted that our RNAseq data was obtained from cBMAT of distal tibiae. It is likely that the fundamental interactions between BMAT and other cells within the marrow niche are similar between regions; however, we do not exclude the possibility that there may be unique interactions between rBMAT and osteoblast/hematopoietic cells.

# Materials and methods
## Resource availability
### Lead contact
Further information and requests for resources and reagents should be directed to and will be fulfilled by the Lead Contact, Ormond A. MacDougald (https://medicine.umich.edu/dept/molecular-integrative-physiology/ormond-macdougald-phd).

### Materials availability
Our *Osterix-FLPo* and FLPo-dependent *Adipoq-Cre* (FAC) mouse models will be available to investigators upon request. All the other data and materials that support the findings of this study are available within the article and supplemental information, or available from the authors upon request.

## Animal
### Generation of BMAd-specific Cre mouse model
To create a BMAd-specific *Cre* mouse model, we expressed mouse codon-optimized FLP (FLPo) (*Raymond et al., 2007*) from the *Osterix* (*Sp7*) locus to recombine and activate a *Cre* expressed from the *Adipoq* locus. After designing sgRNA target sequences against 3'-UTR of endogenous *Osterix* (sgRNA: gatctgagctgggtagaggaagg) and *Adipoq* (sgRNA1: tgaacaagtgagtacacgtgtgg; sgRNA2: cagt gagcagaaaaatagcatgg) genes with the prediction algorithm available at http://crispor.tefor.net, we cloned sgRNA sequences into an expression plasmid bearing both sgRNA scaffold backbone (BB) and Cas9, pSpCas9(BB) (*Ran et al., 2013*), which is also known as pX330 (Addgene plasmid ID: 42230; Watertown, MA). Modified pX330 plasmids were injected into fertilized ova. After culture to the blastocyst stage, Cas9 activity was confirmed by sequencing the predicted cut site. We then inserted a DNA fragment containing an in-frame fusion protein between endogenous *Osterix* and *FLPo*, separated by coding sequence for *P2A* (porcine teschoviral-1) self-cleavage site to allow the full-length proteins to function independently. We also inserted into the 3'-UTR of *Adipoq* an *IRES-F3-Frt-reversed Cre-F3-Frt* cassette with ~1 kb of 5' and 3' flanking homology-arm sequence. The targeting vectors (Cas9 expression plasmid: 5 ng/ul; donor DNA: 10 ng/ul) were injected into fertilized mouse eggs, which were transferred into pseudopregnant recipients to obtain pups. Tail DNA was obtained for genomic PCR and Sanger sequencing to screen F0 founders for desired genetic modifications at the expected locations. These F0 generation chimeric mice were mated with normal mice to obtain mice that are derived exclusively from the modified fertilized eggs. Germline transmitted mice (F1 generation) carrying the designated genetic modifications were bred with reporter mice to validate the activity and efficiency of FLPo and Cre recombinase. These mice were generated by University of Michigan Transgenic Animal Model Core and MDRC Molecular Genetics Core.

### Validation of Osterix-FLPo specificity and efficiency
*Osterix-FLPo* mice were bred with FLP-dependent EGFP reporter mice (derived from https://www.jax.org/strain/012429; Bar Harbor, ME), provided by Dr. David Olson from the University of Michigan MDRC Molecular Genetics Core. Fresh tissue confocal was performed to validate EGFP expression.

### Validation of the specificity and efficiency of FLPo-activated Adipoq-Cre (FAC) mice

Among 80 F0 generation pups, 13 mice carried the FAC cassette. These founder mice were bred with *Osterix-FLPo* mice and mT/mG reporter mice (Stock No. 007676, Jackson Laboratory; Bar Harbor, ME) to further validate Cre recombinase activity and specificity via fresh tissue confocal and immunofluorescent histology. Every single mouse in F1 generation was confirmed and separated for future breeding.

We finally selected F0-#693→ F1-#4376 for BMAd-specific knockout lines. Of note, *Osterix-FLPo* and *FAC* mice were generated on a mixed SJL and C57BL/6J background. *Pnpla2*flox/flox mice purchased from Jackson laboratory (Stock No. 024278; Bar Harbor, ME) were on a C57BL/6J background.

## Animal housing

Mice were housed in a 12 hr light/dark cycle in the Unit for Laboratory Animal Medicine at the University of Michigan, with free access to water. Mice were fed ad libitum or underwent caloric restriction, as indicated. All procedures were approved by the University of Michigan Committee on the Use and Care of Animals with the protocol number as PRO00009687.

## Animal procedures

1. 30% CR. After acclimation to single-housing for 2 weeks, and control diet (D17110202; Research Diets; New Brunswick, NJ) for a week, daily food intake was measured for another week. Mice were then fed a 30% CR diet (D19051601; Research Diets; New Brunswick, NJ) daily at ~6 pm, prior to onset of the dark cycle.
2. Ovariectomy. Female mice at 16 weeks of age underwent ovariectomy (OVX). After 2 weeks recovery, they were either fed an ad libitum or 30% CR diet for 12 weeks.
3. Proximal tibial defect. Surgeries were performed under isoflurane anesthesia, and subcutaneous 0.1 mg/kg buprenorphine was given in 12 h intervals for peri-/post-operative pain management. The proximal tibial defects were obtained by drilling a hole through anterior cortical and trabecular bone, 1–2 mm below the epiphyseal growth plate, with a 0.7 mm low-speed drill.
4. Cold exposure. Mice were single-housed without nesting materials in thermal chambers for 3 weeks at 4°C.
5. Glucose Tolerance Test (GTT). Mice were fasted overnight (16–18 hours). Body weight and fasting glucose levels were measured, followed by an intraperitoneal (i.p.) injection of glucose (1 g glucose/kg body weight). Blood glucose was measured with Bayer Contour test strips at 15, 30, 60, 90, and 120 min time points by cutting the tip of tails.
6. Whole body irradiation. Irradiations were carried out using a Kimtron IC320 (Kimtron Medical; Oxford, CT) at a dose rate of ~4 Gy/min with total dosage of 6 Gy in the University of Michigan Rogel Cancer Center Experimental Irradiation Shared Resource. Dosimetry was carried out using an ionization chamber connected to an electrometer system that is directly traceable to a National Institute of Standards and Technology calibration.

## Fresh tissue confocal microscopy

*Osterix*-driven EGFP expression and BMAd-*Cre*-driven mT/mG reporter mice were sacrificed. Fresh tissues were collected immediately and put in ice-cold PBS, which was protected from light. Soft tissues including adipose tissues and liver were cut into small pieces and placed in a chamber for confocal imaging (Nikon Ti-E Inverted Microscope; Minato City, Tokyo, Japan). Bones were bisected and butterflied on a coverslip for imaging. Both white field and fluorescent images were taken under ×200 magnification.

## Histology and histomorphometry

Tissue histology was performed essentially as described previously (*Li et al., 2019*). Briefly, soft tissues were fixed in formalin, and embedded in paraffin for sectioning. Tibiae were fixed in paraformaldehyde, decalcified in EDTA for at least 2 weeks, and followed by post-decalcification fixation with 4% paraformaldehyde. Bone tissues were then embedded in paraffin, sectioned. After staining with hematoxylin and eosin (H&E), soft tissues were imaged with an Olympus BX51 microscope. Bones were stained with H&E or Tartrate-Resistant Acid Phosphatase (TRAP; Sigma-Aldrich, MO) as indicated, and slides were scanned at ×200 magnification. Static measurements include bone volume fraction, trabecular bone microstructure parameters, osteoblast number and surface, and osteoclast number and eroded surface. Undecalcified tibia was used for plastic sectioning. Mineralized trabecular bone and osteoid were evaluated with Goldner's Trichrome Staining. For dynamic studies, calcein (C0857; Sigma-Aldrich, MO) dissolved in 0.02 g/ml sodium bicarbonate with 0.9% saline at 20 mg/kg was injected intraperitoneally nine- and two-days before sacrifice for quantification of mineral surface (MS), inter-label width (Ir. L. Wi), and mineral apposition rate (MAR) in tibia. Calculations were made

with Bioquant Osteo 2014 (Nashville, TN) software in a blinded manner (*Merceron et al., 2014*; *Morse et al., 2014*).

## µCT analysis

Tibiae were placed in a 19-mm diameter specimen holder and scanned over the entire length of the tibiae using a µCT system (µCT100 Scanco Medical, Bassersdorf, Switzerland). Scan settings were: voxel size 12 µm, 70 kVp, 114 µA, 0.5 mm AL filter, and integration time 500ms. Density measurements were calibrated to the manufacturer's hydroxyapatite phantom. Analysis was performed using the manufacturer's evaluation software with a threshold of 180 for trabecular bone and 280 for cortical bone.

## Marrow fat quantification by osmium tetroxide staining and µCT

After analyses of bone variables, mouse tibiae were decalcified for osmium tetroxide staining, using our previously published method (*Scheller et al., 2015*). In addition, a lower threshold (300 grey-scale units) was used for proximal tibial rBMAT quantification because density of osmium staining is low due to smaller adipocyte size, and with threshold as 400 grey-scale units for cBMAT in distal tibia.

## Immunofluorescent staining

Decalcified tibiae were embedded in OCT compound and used for frozen sectioning at 15 µm. Excess OCT was removed, and tibial tissues were blocked with 10% goat serum for one hour at room temperature. Primary antibodies for GFP (1:500) and RFP (1:200; *Figure 1F*) or ATGL (1:100; *Figure 2B*) were then added to slides and incubated with bone tissues overnight at 4°C. Secondary antibodies (goat anti-Rabbit IgG, Alexa Fluor 594 and goat anti-chicken IgG Alexa Fluor 488 or donkey anti-rabbit IgG (H+L), Alexa Fluor 488) were added to slides following three washes. Two hours later, DAPI staining was performed. Slides were mounted with prolong gold antifade reagent and imaged.

## RNA extraction and quantitative real-time PCR (qPCR)

RNA was extracted from tissues after powdering in liquid nitrogen and lysis in RNAStat60 reagent in a pre-cooled dounce homogenizer. Quantitative PCR was performed using an Applied Biosystems QuantStudio 3 qPCR machine (Waltham, MA). Gene expression was calculated based on a cDNA standard curve within each plate and normalized to expression of the geometric mean of housekeeping genes *Hprt, Rpl32A,* and *Tbp*. qPCR primers are provided in *Supplementary file 3*.

## Immunoblot

Detection of proteins by immunoblot was as described previously (*Mori et al., 2021*).

## Ex vivo lipolysis

Distal tibial BMAT plugs were flushed out from the bone, and four plugs were pooled into one well of a 96-well plate for each n. Pre-warmed 2% BSA HBSS was added to BMAT explants with vehicle (DMSO) or forskolin (FSK, 5 µM), to activate lipolysis (*Litosch et al., 1982*). Cultured media was collected hourly for 4 hr, and glycerol and NEFA concentrations were measured with commercially available kits as listed above in reagents.

## CFU assay

Bone marrow cells of BMAd-*Pnpla2*[+/+] and BMAd-*Pnpla2*[-/-] mice were obtained from femur and tibia. To isolate bone marrow, the bones were flushed with IMDM (Gibco 12440; Waltham, MA) containing penicillin-streptomycin antibiotic (Gibco 15270–063; Waltham, MA). Pelleted cells were counted with a hemocytometer, $1 \times 10^4$ cells were plated in Methocult medium (Stem cell M3534; Vancouver, Canada) in 35 mm culture dishes, and cells were then incubated at 37°C in 5% $CO_2$ with ≥95% humidity for seven days. CFU-G, CFU-M and CFU-GM colonies on each plate were counted using a microscope with a 4X objective lense.

## Bulk RNA sequencing

Distal tibial plugs (cBMAT) from two animals were pooled together for each bulk RNAseq sample. Total RNA was isolated from BMAT for strand specific mRNA sequencing (Beijing Genomics Institute,

China). Over twenty million reads were obtained using a paired-end 100 bp module on DNBSEQ platform. The quality of the raw reads data was checked using FastQC (v.0.11.9) and the filtered reads were aligned to reference genome (UCSC mm10) using STAR with default parameters. All samples passed the post-alignment quality check (QualiMap v.2.2.1). The DEseq2 method was used for differential expression analysis with genotype ($Pnpla2^{+/+}$ vs. $Pnpla2^{-/-}$) and treatment (CR vs. ad libitum) as the main effects. Gene ontology analysis was done using MetaScape. The Principal Component Analysis (PCA) plot was generated using the PlotPCA function building in DESeq2 package. To compare the number of differentially expressed genes between genotype/treatment groups, volcano plots were constructed using the Enhanced Volcano package under R environment. Heatmap plots were generated using pheatmap package under R environment and the complete-linkage clustering method was used for the hierarchical cluster of genes. These data are available through NCBI GEO with the following accession number, GSE183784.

## Flow cytometry

Femurs were isolated from mice. Bone marrow was harvested by flushing the femurs with 1 mL of ice-cold PEB (1 X PBS with 2 mM EDTA and 0.5% bovine serum albumin). Red blood cells were lysed once by adding 1 mL of RBC Lysis Buffer (155 mM $NH_4Cl$, 10 mM $KHCO_3$, 0.1 mM EDTA) and gently pipetting to mix. Cells were immediately pelleted by centrifugation and resuspended in 1 mL of ice-cold PEB. Cells were stained for 30 min in PEB buffer with the indicated antibodies below and analyzed on the BD LSRFortessa or BD FACSAria III. Data was analyzed using FlowJo software (BD Biosciences, version 10.8). Dead cells and doublets were excluded based on FSC and SSC distribution. To stain for mature leukocytes antibodies used were against CD45, Ly6G, CD11b, CD115, CD19, and CD3e. All $CD45^+$ cells were gated first for further identification. Neutrophils were defined as $Ly6G^+CD11b^+$, monocytes were defined as $Ly6G^-CD11b^+CD115^+$, B cells were defined as $Ly6G^-CD11b^-CD19^+$, and T cells were defined as $Ly6G^-CD11b^-CD3e^+$. To stain for bone marrow neutrophil populations, antibodies used were against Ly6G, CD11b, cKit, CXCR2, CXCR4, and CD62L. Pre-neutrophils were defined as $Ly6G^+CD11b^+cKit^+$, immature neutrophils were defined as $Ly6G^+CD11b^+cKit^-CXCR2^{lo}$, and mature neutrophils were defined as $Ly6G^+CD11b^+cKit^-CXCR2^{hi}$. To stain for hematopoietic stem and progenitor cells (HSPCs) antibodies used were against a lineage panel (Gr-1, CD11b, B220, CD3e, TER119), cKit, Sca-1, CD150, CD48, CD105, and CD16/32. After gating on the lineage⁻ population, HSPCs were defined as follows; HSCs as $LSKCD150^+CD48^-$, MPPs as $LSKCD150^-CD48^-$, HPC1 as $LSKCD150^-CD48^+$, HPC2 as $LSKCD150^+CD48^+$, GMPs as $LKCD150^-CD16/32^+$, PreGMs as $LKCD150^-CD105^-$, PreMegEs as $LKCD150^+CD105^-$, and PreCFUe as $LKCD150^+CD105^+$. LSK = lineage⁻Sca1⁺cKit⁺, LK = lineage⁻cKit⁺.

## Bone regeneration analysis

Nine days after the proximal tibial defect surgery, proximal tibiae were collected for μCT scanning. After construction, new generated trabecular and cortical bone were quantified by Dragonfly software (Montréal, Canada). Under the full view of 3D and 2D images, a cylinder-shaped region of interest (ROI) was defined in trabecular or cortical bone defect area (as shown in *Figure 6A*) with 3D dimension as 0.3 mm (diameter) x 0.7 mm (height) for trabecular compartment and 0.3 mm (diameter) x 0.2 mm (height) for cortical compartment. An automatic split at otsu threshold for each bone was collect first, and then mean threshold for a whole cohort was calculated. This average threshold was applied to each bone to normalize bone volume fraction and mineral content.

## Statistics

We calculated the minimal animal number required for studies based on the mean and SD values to make sure we had adequate animals per group to address our hypothesis. All the mice were randomly assigned to the indicated groups. Although the investigators responsible for group allocation were not blinded to the allocation scheme, they were blinded to group allocation during data collection, and the investigators responsible for analyses were blinded to the allocation scheme.

Significant differences between groups were assessed using a two-sample *t*-test or ANOVA with post-tests as appropriate: one-way ANOVA with Tukey's multiple comparisons test, two-way ANOVA with Sidak's multiple comparisons test, and three-way ANOVA analysis as appropriate. All analyses were conducted using the GraphPad Prism version 9. All graphical presentations are mean +/-SD. For

statistical comparisons, a p-value of $<0.05$ was considered significant. All experiments were repeated at least twice.

## Acknowledgements

This work was supported by grants or fellowships from the NIH to OAM (R01 DK62876; R24 DK092759; R01 DK126230; R01 AG069795), SMR (T32 GM835326; F31 DK12272301), DPB (T32 HD007505; T32 GM007863), KS (R01DK115583), KTL (T32 DK071212; F32 DK122654), RLS (T32 DK101357; F32 DK123887), KDH (R01AR066028), and CJR (R24DK092759). ZL was supported by a fellowship from the American Diabetes Association (1–18-PDF-087) and EB from the American Heart Association (20-PAF00361). This research was also supported by a Pilot & Feasibility grant and core facilities of the Michigan Integrative Musculoskeletal Health Core Center (P30 AR069620), Michigan Diabetes Research Center (P30 DK020572), Michigan Nutrition and Obesity Center (P30 DK089503), and University of Michigan Comprehensive Cancer Center (P30 CA046592).

## Additional information

### Funding

| Funder | Grant reference number | Author |
| --- | --- | --- |
| National Institutes of Health | R01 DK62876 | Ormond A MacDougald |
| National Institutes of Health | R24 DK092759 | Ormond A MacDougald |
| National Institutes of Health | R01 DK126230 | Ormond A MacDougald |
| National Institutes of Health | R01 AG069795 | Ormond A MacDougald |
| National Institutes of Health | T32 GM835326 | Steven M Romanelli |
| National Institutes of Health | F31 DK12272301 | Steven M Romanelli |
| National Institutes of Health | T32 HD007505 | Devika P Bagchi |
| National Institutes of Health | T32 GM007863 | Devika P Bagchi |
| National Institutes of Health | R01DK115583 | Kanakadurga Singer |
| National Institutes of Health | T32 DK071212 | Kenneth T Lewis |
| National Institutes of Health | F32 DK122654 | Kenneth T Lewis |
| National Institutes of Health | T32 DK101357 | Rebecca L Schill |
| National Institutes of Health | F32 DK123887 | Rebecca L Schill |
| National Institutes of Health | R01AR066028 | Kurt D Hankenson |
| National Institutes of Health | R24DK092759 | Clifford J Rosen |
| American Diabetes Association | 1-18-PDF-087 | Ziru Li |

| Funder | Grant reference number | Author |
|---|---|---|
| American Heart Association | 20-PAF00361 | Emily Bowers |

The funders had no role in study design, data collection and interpretation, or the decision to submit the work for publication.

## Author contributions

Ziru Li, Conceptualization, Data curation, Formal analysis, Investigation, Methodology, Project administration, Software, Validation, Visualization, Writing - original draft; Emily Bowers, Data curation, Formal analysis, Investigation, Methodology, Resources, Software, Visualization; Junxiong Zhu, Data curation, Investigation, Methodology, Resources, Visualization; Hui Yu, Data curation, Formal analysis, Methodology, Resources, Software, Visualization; Julie Hardij, Data curation, Investigation, Visualization; Devika P Bagchi, Hiroyuki Mori, Kenneth T Lewis, Katrina Granger, Rebecca L Schill, Steven M Romanelli, Investigation; Simin Abrishami, Data curation, Methodology; Kurt D Hankenson, Kanakadurga Singer, Conceptualization, Funding acquisition, Methodology, Resources, Supervision, Writing – review and editing; Clifford J Rosen, Conceptualization, Funding acquisition, Project administration, Resources, Supervision, Writing – review and editing; Ormond A MacDougald, Conceptualization, Data curation, Funding acquisition, Project administration, Resources, Supervision, Writing – review and editing

## Author ORCIDs

Ziru Li http://orcid.org/0000-0002-2755-9299
Hui Yu http://orcid.org/0000-0001-5249-0193
Kanakadurga Singer http://orcid.org/0000-0001-8278-3800
Clifford J Rosen http://orcid.org/0000-0003-3436-8199
Ormond A MacDougald http://orcid.org/0000-0001-6907-7960

## Ethics

All procedures were approved by the University of Michigan Committee on the Use and Care of Animals with the protocol number as PRO00009687.

## Decision letter and Author response

Decision letter https://doi.org/10.7554/eLife.78496.sa1
Author response https://doi.org/10.7554/eLife.78496.sa2

# Additional files

## Supplementary files

• Supplementary file 1. BMAd-lipolysis deficiency has subtle effects on circulating mature blood cells. Whole blood from male mice at 24 weeks of age fed ad libitum (top) or a 30% CR diet for 6weeks (bottom) was submitted for complete blood counts (CBC). White- and red- blood cell related parameters are shown. Multiple unpaired t tests had been performed crossing all parameters, *P* values were adjusted for multiple comparisons using Two-stage step-up (Benjamini, Krieger, and Yekutieli) with FDR method.

• Supplementary file 2. BMAd-lipolysis deficiency causes mild changes in bone marrow hematopoietic cells. Femoral bone marrow cells from male mice at 24 weeks of age fed ad libitum (top) or a 30% CR diet for 6 weeks (bottom) were collected and stained with antibodies for flow cytometry analyses. Mature blood cells and hematopoietic stem/progenitor cells (HSPCs) were counted. Multiple unpaired t tests had been performed crossing all parameters, *P* values were adjusted for multiple comparisons using Two-stage step-up (Benjamini, Krieger, and Yekutieli) with FDR method.

• Supplementary file 3. PCR primer list.

• MDAR checklist

## Data availability

The accession number for the BMAT bulk RNA seq data reported in this paper is GEO: GSE183784.

The following dataset was generated:

| Author(s) | Year | Dataset title | Dataset URL | Database and Identifier |
|---|---|---|---|---|
| Li Z, Yu H, MacDougald O | 2021 | Bone marrow adipocyte (BMAd)-specific deletion of Pnpla2 reveals local roles for lipolysis to fuel cells of bone and the marrow niche | https://www.ncbi. nlm.nih.gov/geo/ query/acc.cgi?acc= GSE183784 | NCBI Gene Expression Omnibus, GSE183784 |

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

## Appendix 1

**Appendix 1—key resources table**

| Reagent type (species) or resource | Designation | Source or reference | Identifiers | Additional information |
|---|---|---|---|---|
| Strain, strain background (*Mus musculus*, SJLxC57BL/6J) | *Osterix-FLPo* | Generated in University of Michigan Transgenic Animal Model Core | Donated to JAX: Stock No. 037208 | Male and female |
| Strain, strain background (*Mus musculus*, SJLxC57BL/6J) | FLPo-dependent A*dipoq-Cre* (FAC) | Generated in University of Michigan Transgenic Animal Model Core | N/A | Male and female |
| Strain, strain background (*Mus musculus*, SJLxC57BL/6J) | Frt-floxed EGFP | Generated in University of Michigan Transgenic Animal Model Core | N/A | Male and female |
| Strain, strain background (*Mus musculus*, C57BL/6J) | mT/mG | Jackson Laboratory | Stock No. 007676 | Male and female |
| Strain, strain background (*Mus musculus*, C57BL/6J) | *Pnpla2*$^{flox/flox}$ | Jackson Laboratory | Stock No. 024278 | Male and female |
| Antibody | Anti-Adiponectin (rabbit monoclonal) | Sigma-Aldrich | A6354 | WB (1:1000) |
| Antibody | Anti-FABP4/A-FABP Antibody (goat polyclonal) | R&D Systems | AF1443 | WB (1:1000) |
| Antibody | Anti-alpha Tubulin Antibody (rat monoclonal) | Thermo Fisher Scientific | MA180017 | WB (1:1000) |
| Antibody | Anti-ERK 2 Antibody (mouse monoclonal) | Santa Cruz Biotechnology | sc-1647 | WB (1:1000) |
| Antibody | Anti-Albumin antibody (rabbit monoclonal) | Abcam | Ab207327 | WB (1:1000) |
| Antibody | Anti-ATGL Antibody (rabbit polyclonal) | Cell Signaling Technology | 2138S | WB (1:1000) |
| Antibody | Anti-ATGL Antibody (rabbit monoclonal) | Abcam | ab207799 | IF (1:100) |
| Antibody | Anti-GFP antibody (chicken polyclonal) | Abcam | ab13970 | IF (1:500) |
| Antibody | Anti-RFP Antibody Pre-adsorbed (rabbit polyclonal) | Rockland | 600-401-379 | IF (1:200) |
| Antibody | Goat anti Rabbit IgG (H+L) Secondary Antibody, Alexa Fluor 594 (goat polyclonal) | Invitrogen | A11012 | IF (1:100) |
| Antibody | Alexa Fluor 488 goat anti-chicken IgG (H+L) (goat polyclonal) | Invitrogen | A11039 | IF (1:100) |

*Appendix 1 Continued on next page*

*Appendix 1 Continued*

| Reagent type (species) or resource | Designation | Source or reference | Identifiers | Additional information |
|---|---|---|---|---|
| Antibody | Anti-Ly6G FITC (rat monoclonal) | BD Biosciences | 551460 | FACS (1:200) |
| Antibody | Anti-CD11b APC (rat monoclonal) | Invitrogen | 17-0112-82 | FACS (1:200) |
| Antibody | Anti-CD115 APC-Cy7 (rat monoclonal) | Biolegend | 135531 | FACS (1:100) |
| Antibody | Anti-CD3e PE-Cy7 (Armenian Hamster monoclonal) | Biolegend | 100319 | FACS (1:200) |
| Antibody | Anti-CD19 Pacific Blue (rat monoclonal) | Invitrogen | 48-0193-82 | FACS (1:200) |
| Antibody | Anti-CD45 AlexaFluor700 (rat monoclonal) | BD Biosciences | 560510 | FACS (1:200) |
| Antibody | Anti-Gr-1 Biotin (rat monoclonal) | Biolegend | 79750 | FACS (1:200) |
| Antibody | Anti-CD11b Biotin (rat monoclonal) | Biolegend | 79749 | FACS (1:200) |
| Antibody | Anti-B220 Biotin (rat monoclonal) | Biolegend | 79752 | FACS (1:200) |
| Antibody | Anti-CD3e Biotin (Armenian Hamster monoclonal) | Biolegend | 79751 | FACS (1:200) |
| Antibody | Anti-TER119 Biotin (Armenian Hamster monoclonal) | Biolegend | 79748 | FACS (1:200) |
| Antibody | Anti-Sca1 PE-Cy7 (rat monoclonal) | Invitrogen | 25-5981-82 | FACS (1:200) |
| Antibody | Anti-cKit (CD117) APC-Cy7 (rat monoclonal) | Biolegend | 105826 | FACS (1:100) |
| Antibody | Anti-CD150 BrilliantViolet 421 (rat monoclonal) | Biolegend | 115925 | FACS (1:200) |
| Antibody | Anti-CD48 FITC (Armenian Hamster monoclonal) | Invitrogen | 11-0481-85 | FACS (1:100) |
| Antibody | Anti-CD16/32 PerCP-Cy5.5 (rat monoclonal) | Biolegend | 101324 | FACS (1:200) |
| Antibody | Anti-CD105 APC (rat monoclonal) | Biolegend | 120413 | FACS (1:200) |
| Antibody | Anti-CXCR2 PE (rat monoclonal) | Biolegend | 149304 | FACS (1:200) |
| Antibody | Anti-CXCR4 PE-Dazzle (rat monoclonal) | Biolegend | 146514 | FACS (1:100) |
| Antibody | Anti-CD62L BrilliantViolet 421 (rat monoclonal) | Biolegend | 104435 | FACS (1:200) |

*Appendix 1 Continued on next page*

*Appendix 1 Continued*

| Reagent type (species) or resource | Designation | Source or reference | Identifiers | Additional information |
|---|---|---|---|---|
| Commercial assay or kit | Acid Phosphatase Leukocyte (TRAP) Kit | Sigma-Aldrich | 387A-1KT | |
| Commercial assay or kit | Free Glycerol Determination Kit | Sigma-Aldrich | FG0100 | |
| Commercial assay or kit | NEFA Reagent (NEFA-HR(2)) | FUJIFILM Wako Diagnostics | NC9517309 | |
| Commercial assay or kit | BCA Protein Assay Kit | Thermo Fisher Scientific | 23225 | |
| Commercial assay or kit | RAT/MOUSE P1NP ELISA KIT | Immunodiagnostic Systems Inc | NC9666468 | |
| Commercial assay or kit | Mouse TRANCE/RANK L/TNFSF11 Quantikine ELISA Kit | R&D Systems | MTR00 | |
| Commercial assay or kit | MOUSE TRAP ASSAY | Immunodiagnostic Systems Inc | NC9360739 | |
| Commercial assay or kit | MethoCult GF M3534 | ATEMCELL | 03534 | |
| Commercial assay or kit | DNA-free Kit | Life Technologies | AM1906 | |
| Chemical compound, drug | Calcein | Sigma-Aldrich | C0875 | |
| Chemical compound, drug | EDTA | DOT Scientific Inc | dse57020 | |
| Chemical compound, drug | Tetroxide Osmium | Electron Microscopy Sciences | 19190 | |
| Chemical compound, drug | Forskolin | Cayman Chemical Company | 11018 | |
| Chemical compound, drug | Bovine Serum Albumin (BSA), Fraction V | Gold Biotechnology | A-421–250 | |
| Chemical compound, drug | qPCRBio SyGreen Mix Hi-ROX Blue | Innovative Solutions | 4SPB20.16 | |
| Chemical compound, drug | PCRBio HS Taq Mix Red | Innovative Solutions | 4SPB10.23 | |
| Chemical compound, drug | Agarose | Thermo Fisher Scientific | BP160-500 | |
| Chemical compound, drug | RNA STAT-60 | AMSBIO | CS-502 | |
| Chemical compound, drug | M-MLV Reverse Transcriptase | Thermo Fisher Scientific | 28025013 | |
| Chemical compound, drug | 100bp DNA Ladder | NEB | N3231S | |
| Software, algorithm | Microsoft Office | Microsoft | https://its.umich.edu/communication/collaboration/microsoft-office-365/getting-started | |

*Appendix 1 Continued on next page*

*Appendix 1 Continued*

| Reagent type (species) or resource | Designation | Source or reference | Identifiers | Additional information |
|---|---|---|---|---|
| Software, algorithm | Adobe photoshop | Adobe | https://www.adobe.com/creativecloud/desktop-app.html | |
| Software, algorithm | Prism 9 | GraphPad software | https://www.graphpad.com/ | |
| Software, algorithm | Image J | Image J | https://imagej.nih.gov/ij/ | |
| Software, algorithm | MetaMorph | BioVision Technologies | https://www.biovis.com/metamorph.html | |
| Software, algorithm | Scano µCT 100 | SCANCO Medical AG | https://www.scanco.ch/ | |
| Software, algorithm | BIOQUANT OSTEO | BIOQUANT Image analysis corporation | https://bioquant.com/ | |
| Software, algorithm | STAR | PMID:23104886 | https://github.com/alexdobin/STAR; *Dobin et al., 2013*; *Dobin, 2022* | |
| Software, algorithm | DESeq2 | PMID:25516281 | https://bioconductor.org/packages/release/bioc/html/DESeq2.html | |
| Software, algorithm | QualiMap-2 | PMID:26428292 | https://qualimap.conesalab.org | |
| Software, algorithm | R | The R Foundation | https://www.r-project.org/ | |
| Software, algorithm | Metascape | PMID:30944313 | https://metascape.org/gp/index.html | |
| Software, algorithm | Dragonfly | ORS - OBJECT RESEARCH SYSTEMS | https://www.theobjects.com/dragonfly/index.html | |
| Software, algorithm | FlowJo | BD Biosciences | https://www.flowjo.com/solutions/flowjo | |
| Other | Streptavidin BrilliantViolet 510 | Biolegend | 405233 | FACS (1:200) |
| Other | Element HT5 Veterinary Hematology Analyzer | Heska | https://www.heska.com/product/element-ht5/ | Service provided by UMICH In-Vivo Animal Core (IVAC) |
| Other | FACSAria III cell sorter | BD Biosciences | N/A | Shared equipment in UMICH Flow Cytometry Core |
| Other | LSRFortessa Cell Analyzer | BD Biosciences | N/A | Shared equipment in UMICH Flow Cytometry Core |
| Other | Nikon A1 Confocal Microscope | Nikon | N/A | Shared equipment in UMICH Microscopy and Imaging Analysis Core (MIAC) Michigan Diabetes Research Center |

*Appendix 1 Continued*

| Reagent type (species) or resource | Designation | Source or reference | Identifiers | Additional information |
|---|---|---|---|---|
| Other | Bayer Contour Next Test Glucose Strips | Diabetic Corner | ByrContournext | https://www.contournextone.com/siteassets/pdf/web85688006_cntrnxtone_ug_r01-17.pdf |
| Other | Scanco µCT 100 micro-computed tomography system | SCANCO Medical AG | https://www.scanco.ch/ | Service provided by UMICH School of Dentistry MicroCT Core |
| Other | Olympus BX51 | Olympus | N/A | https://www.olympus-lifescience.com/en/microscope-resource/primer/techniques/fluorescence/bx51fluorescence/ |
| Other | StepOnePlus Real-Time PCR System | Thermo Fisher Scientific | 4376600 | https://www.thermofisher.com/order/catalog/product/4376600 |
| Other | Microtome | Leica | RM2235 | https://www.leicabiosystems.com/en-br/histology-equipment/microtomes/leica-rm2235/ |
| Other | Slide scanner | Nikon | N/A | Shared equipment in UMICH Orthopaedic Research Laboratories (ORL) Histology Core |

