## [Editor Report]

Through the cell selective deletion in bone marrow marrow adipocytes (BMAds) of an enzyme for lipolysis, the authors demonstrate elegantly a role for this pathway in the maintenance of hematopoiesis and bone mass during periods of caloric stress, injury and energy restriction. This study establishes for the first time a role for the bone marrow adipocyte in energy balance.

---

## [Decision Letter]

**Decision letter after peer review:**

Thank you for submitting your article "Lipolysis of bone marrow adipocytes is required to fuel bone and the marrow niche during energy deficits" for consideration by *eLife*. Your article has been reviewed by 3 peer reviewers, and the evaluation has been overseen by me. The following individual involved in the review of your submission has agreed to reveal their identity: William P Cawthorn (Reviewer #3).

In view of our discussions as a consultative review group, I have drafted this to help you prepare a revised submission.

Essential revisions:

Overall, the work is of high quality. The development of the new mouse model is a major strength that could be transformative for mechanistic studies of BMAd formation and function.

1. A main requirement is to measure circulating adiponectin in experimental mice. If the KO CR mice have lower adiponectin than the WT CR mice then this might be confounding the phenotypes observed. This will need discussion.

2. Many of the current figures and data analysis should also be updated to address limitations mentioned in the reviewer comments (below).

3. Update the statistical analysis so that the authors are not missing genotype*diet interactions, and present the AL and CR data side by side in all relevant figures (e.g. Figure 3, Figure 3 —figure supplement 1, Figure 3 —figure supplement 2). This should be easy to do, though the updated analyses might lead to updated conclusions.

4. Further analyze transcriptomic data for potential clues about the mechanisms underlying the hematological phenotypes (as they've done currently for the bone phenotype). Also highlight the limitation that the transcriptomic data are from distal tibia, whereas the bone analyses and hematological analyses are largely from other regions of the skeleton.

5. Update the OVX analyses to compare the OVX mice with ovary-intact females (as done for uterus masses in Figure 3 —figure supplement 4D). This would partially address the limitation, highlighted by Reviewer 2, that the authors haven't included appropriate sham or estrogen-replete controls for OVX mice. This limitation should be mentioned in the Discussion. Alternatively, the OVX experiment could be removed from the paper, as it is not particularly adding much.

6. Need a proper description and discussion of the bone injury healing experiment.

7. Specific comments under Recommendations to Authors need to be addressed in a point-to-point rebuttal.

*Reviewer #1 (Recommendations for the authors):*

Overall this is a good manuscript and no major revisions are recommended. However, there are several grammatical errors (e.g. line 521) that would require a careful proofreading to improve. Moreover, given the general readership of *eLife*, expansion of the introduction to include more background on topics that as written are fairly field-specific would certainly enhance the overall appeal to a wider audience.

*Reviewer #2 (Recommendations for the authors):*

1) Consider the mechanism by which ATGL deficiency induces lipogenesis with respect to the following. There is a substantial accumulation of marrow fat in Pnpla2 -/- females under CR conditions which supersedes an amount of fat in control mice in the same conditions. This indicates massive lipogenesis rather than BMAd acting as a sink to circulating FA, as the authors propose.

2) To assess whether estrogen protects female Pnpla2 -/- mice from negative effects of CR, mice were OVX. Surprisingly, OVX control and Pnpla2 -/- females "gained" bone when exposed to CR. There is no plausible explanation for this effect, which is in striking contrast to the CR effect in males. In addition, Sham-operated control is not provided in this experiment which poses the question of whether it is a bone gain or protection from bone loss. The conclusion that "metabolic benefits of CR diet combat detrimental effects of estrogen deficiency and/or aging" is overstated and applies only to females. This is an intriguing observation but needs more insight. What is its relevance to humans? Is there any evidence that malnourished postmenopausal women are gaining bone?

3) Pnpla2 -/- male but not female mice have mildly affected myelopoiesis which is exacerbated in CR conditions. CR also affected hematopoietic marrow regeneration after irradiation which correlated with decreased number of hematopoietic stem cells. There is no evidence that these alterations are due to fuel deficiency in the marrow environment. The possibility that there are changes in the signaling milieu should be tested experimentally or considered based on transcriptomic data.

4) In regard to the bone regeneration model there is some misunderstanding. What the authors consider as trabecular bone is actually woven bone that is removed during the final stages of healing which occur at week 4 after injury. Referring to it as trabecular bone is not correct because the woven bone has a different origin and structure. In this model, day 9 after injury correlates with initiation of mineralization which robustness depends on previously formed hematoma and callus, and penetration of new vessels. It is possible that Pnpla2 deficiency affects early stages of bone healing not necessarily osteoblast bone-forming activity. Again, the transcriptomic data may give some clue, although they are done on marrow fat isolated from the distal tibia which is considered metabolically different than marrow fat in the proximal tibia where the bone defect was created.

*Reviewer #3 (Recommendations for the authors):*

General points:

a. Adiponectin deficiency in the Pnpla2 WT vs KO mice:

Please analyse circulating adiponectin in the Pnpla2 WT vs KO mice. This is important because you show that the BMAd-Cre mice have lower adiponectin and that the KO CR mice do not increase BMAT adiponectin expression to the same extent as the WT CR mice (i.e. Figure 5D and Figure 5 supplemental figure 2B) Notably, adiponectin KO mice have been shown to resist bone loss and BMAT expansion during CR (Zhu et al., 2020). Therefore, if the KO CR mice do have lower adiponectin than the WT CR mice then this might be influencing their bone phenotype. It would also be helpful to know if the BMAd-Cre mice do increase circulating adiponectin during CR, despite having lower adiponectin expression than the non-Cre mice.

Because the hypoadiponectinaemia may be confounding, it would be helpful to compare the BMAd-Pnpla2+/+ and BMAd-Pnpla2-/- mice with Cre-negative Pnpla2-fl/fl mice, both under AL and CR conditions.

Related to these considerations, in lines 576-578 you mention a few previous studies of adiponectin and bone, but the Zhu 2020 paper about adiponectin KO and CR is not cited. Please update this part of the discussion to mention this very relevant paper.

b. Analysis of diet and diet-genotype effects:

i. In Figure 3 and its supplemental figures 1 and 2 please can the data from ad libitum (AL) mice be analysed together with the data from the calorie-restricted (CR) mice? Currently the two diets are assessed separately and therefore it's not possible to detect effects of CR or if the BMAd Pnpla2 KO significantly alters these diet effects. Knowing this would be extremely informative as it may be that, while the KO alone has no effect within each diet, it might alter the CR effects. For some readouts this may also increase statistical power to detect significant effects of diet or genotype alone.

ii. Related to this, in Figure 3 (supplemental figures 3 and 4), Figure 4 (and its two supplemental figures), Figure 5 (supplemental figure 2B) and Figure 6 the four genotype-diet groups are analysed side-by-side, but they are compared using 1-way ANOVA. For these figures using a 2-way (or sometimes 3-way) ANOVA is more appropriate and, as above, is required if you are to detect diet effects and genotype-diet interactions. For example, the time course data in Figure 4 supplemental figure 1C-E should be assessed by 3-way ANOVA with genotype, diet and time as the independent variables, while the data in Figure 4 and panels F-G of the supplemental figure should be assessed by 2-way ANOVA with genotype and diet as the independent variables. As above, this is important because you can then test for interactions between the variables, which might reveal KO effects that cannot be detected by 1-way ANOVA. In some cases, the 2-way ANOVA is also required to support existing statements in the results. For example, Line 303 ("…mature neutrophils are affected only by CR (Figure 4H)") and Lines 307-308 ("fewer granulocyte progenitor colonies (CFU-G) from CR BMAd-Pnpla2-/- mice (Figure 4I)") are not supported by any statistical analysis.

c. In the Results section the findings frequently are described in the present tense. Convention is to report the results in the past tense (i.e. what was found) and then, in the Discussion, to interpret these data in the present tense (i.e. what the results show/reveal etc). Please can this be updated throughout the Results section?

d. Genotypes of BMAd-Cre mice:

i. In Supplemental Figure 2, you distinguish homozygous vs heterozygous BMAd-Cre mice. I think it would help the reader to clarify that this relates to the FAC transgene, but it's not clear if these mice have one or two alleles of the Osterix-FLPo transgene. Please can you clarify in the text of the results

ii. Lines 174-175: You state that mice used for the Pnpla2 KO studies were positive for both Osterix-FLPo and FAC; were they homozygous for both transgenes or was one of the heterozygous (etc)? This needs to be clarified in the text.

Specific points:

e. Figure 1 —figure supplement 1:

i. For 1C I'm confused by the genotype labelling for the Osterix mouse above the blot: does the '+' refer to the presence of FLPo? It's confusing as usually '+' would refer to the WT allele; please clarify in the figure and the legend.

ii. In the main text (lines 126-128) you state, "Cre in the correct orientation… is only observed in caudal vertebral DNA of mice positive for at least one copy each of Osterix (Mut band) and FAC (Ori band)". This is shown in Supp Figure 1F so please refer to this figure after making this statement in the text. Also, it is logical to move Supp Figure 1F to Supp Figure 1D, as it follows immediately from 1C.

f. Lines 144-145, "Indeed, Cre efficiency is ~80% in both male and female mice over 16 weeks of age with one Cre allele, and over 90% at 12 weeks of age in mice with two Cre alleles". Please explain how you calculated these percentages of efficiency; no quantification is shown in the figures.

g. Lines 160-161: Please can you reword this text for clarity, e.g. "We found that insertion of the IRES-Cre cassette caused a small, non-significant decrease in Adipoq mRNA expression in whole caudal vertebrae or distal tibiae (Figure Supp 2A), whereas Adipoq mRNA in WAT…"

h. Line 166: The greater decrease in circulating adiponectin is clear from the data. Therefore, please revise the wording to "Circulating adiponectin concentrations were decreased even further…".

i. Figure 1 —figure supplement 2F: please can you refer to this in the figure legend, e.g. "(F) High, medium and low molecular weight forms of adiponectin were quantified by Image J."

j. Figure 2

i. 2B: There is still some ATGL fluorescence in the proximal tibia of the KO mice. Where is this coming from? Perhaps ATGL is expressed earlier than adiponectin in BMAd development, so some differentiating BMAds may still have intact ATGL expression?

ii. 2D-F: The differences look greatest in the proximal tibial diaphysis, above the tibia-fibula junction. Is there a reason that this region was not quantified by osmium?

k. Lines 220-222, "These data also indicate that expansion of BMAT is not sufficient to cause bone loss and that the correlation between elevated fracture risk and expansion of BMAT under a variety of clinical situations is not necessarily a causal relationship." This is an interesting and important point but is better suited to the Discussion section.

l. Figure 3 (main figure and Supplemental figures):

i. Figure 3 Supp 1B vs 1K: Was glucose tolerance improved by CR? Fasting and peak glucose in the CR mice (1K) look similar to in the ad libitum mice (1B). It would be informative to calculate the AUC for the GTTs and to then plot this for all four groups on the same axis, comparing by 2-way ANOVA.

ii. In Figure 3 Supp 1O you show nicely that total BMAT volume is similar between KO and control during CR. Is BMAd size the same? i.e. do the KO mice still have larger BMAds than control mice during CR?

iii. Lines 243-244: You state "we observed a trend (P = 0.07) towards reduced NEFA concentrations in bone marrow supernatant of CR BMAd-Pnpla2-/- mice". However, the same trend seems to occur for the AL mice (Supplemental 3I). Please can you add P values to Figure 3 Supp 1I so that the AL data can be compared more directly with glycerol and NEFA during CR (Figure 3 Supp 1R)? Better still would be to have all four groups compared on a single axis (as suggested above in General Point b), as this may give you sufficient power to detect a significant genotype effect on NEFA in the BM supernatant.

iv. In line 255 you state that TRACP5b is decreased but from the figure this isn't significant, so please temper this statement in the main text. Alternatively, if you were to analyse the AL and CR mice together by 2-way ANOVA you might then have enough power to detect a significant decrease in the KO mice, irrespective of diet.

m. Was DAPI used for live/dead staining in your flow cytometry? This is not absolutely essential but should be stated in the Methods section if this was or was not done.

n. Table 2: Please confirm that these P values are adjusted for multiple comparisons and update the Table legend to state how statistical significance was determined.

o. Figure 3 —figure supplement 3:

i. BM adiposity is shown histologically in panel C but this is not quantified, so it's not clear to what extent this is altered by genotype and/or diet. Please can you quantify BM adiposity from these images? Even better would be to use osmium staining so that the female data could be compared with the male data in Figure 3 supplement 1.

p. In Figure 5B-C and the related supplemental figures, what's the rationale for comparing CR KO mice to AL WT mice? I think this comparison is invalid because it is being made across two independent variables. For example, if the target gene expression is similar in CR WT vs CR KO, but (e.g.) lower in AL WT vs AL KO, then the fold change vs AL WT will be similar for CR WT and CR KO, but the fold change for CR KO vs AL KO will be greater. To my mind, this would mean that the KO is altering the effect of CR, but your approach would not detect this. So, if the goal is to identify genes regulated by CR irrespective of Pnpla2 deficiency then this should be done by comparing the effects of CR within each genotype.

q. The finding of impaired bone regeneration with CR and Pnpla2 KO is consistent with your conclusion, i.e. that BMAd lipolysis is important for bone regeneration. However, another interpretation relates to the findings of Ambrosi 2017, who show that BMAds impair fracture healing (Ambrosi et al., 2017). Their data suggest that BMAds can secrete DPP4 and that this impairs bone healing. Therefore, another possibility for your observation is that increased BMAT in the Pnpla2 KO mice results in increased production of DPP4 in the bone marrow, which impairs bone regeneration. Please can you measure DPP4 expression within the BM and/or discuss this possibility in the Discussion section?

r. Line 453: Please correct 'Figure 4I' to 'Figure 4H'. Also, you can't really state that BMD declines when this effect is not statistically significant; can you test this further (e.g. use 2-way ANOVA and/or in a second cohort of mice), or at least temper the language used to describe this effect?

s. You show that myelopoiesis is impaired by BMAd-specific Pnpla2 deficiency, but do not go into the mechanism about why only the myeloid system, and not the CLP or HSC (which are also in the bone marrow), is dependent on energy supply from BMAds under CR. Please can you speculate on this in the Discussion?

References

Ambrosi, T.H., Scialdone, A., Graja, A., Gohlke, S., Jank, A.M., Bocian, C., Woelk, L., Fan, H., Logan, D.W., Schurmann, A., et al. (2017). Adipocyte Accumulation in the Bone Marrow during Obesity and Aging Impairs Stem Cell-Based Hematopoietic and Bone Regeneration. Cell Stem Cell 20, 771-784 e776. 10.1016/j.stem.2017.02.009.

Zhu, J., Liu, C., Jia, J., Zhang, C., Yuan, W., Leng, H., Xu, Y., and Song, C. (2020). Short-term caloric restriction induced bone loss in both axial and appendicular bones by increasing adiponectin. Ann. N. Y. Acad. Sci. n/a. 10.1111/nyas.14380.

---

## [Author Response]

Essential revisions:Overall, the work is of high quality. The development of the new mouse model is a major strength that could be transformative for mechanistic studies of BMAd formation and function.1. A main requirement is to measure circulating adiponectin in experimental mice. If the KO CR mice have lower adiponectin than the WT CR mice then this might be confounding the phenotypes observed. This will need discussion.

We have measured circulating adiponectin concentrations and reported the data in Figure 3 —figure supplement 1V. Although BMAd-Cre mice have hypoadiponectinemia, they still respond to caloric restriction by increasing circulating adiponectin concentrations. Circulating concentrations of adiponectin at baseline and with CR are similar between BMAd-Pnpla2^+/+^ and BMAd-Pnpla2^-/-^ mice.

2. Many of the current figures and data analysis should also be updated to address limitations mentioned in the reviewer comments (below).

We have gone through all the datasets and finished the multiple comparisons with FDR method for Figure 3, Tables 1 and 2; and two-way or three-way ANOVA analyses when appropriate for other figures.

3. Update the statistical analysis so that the authors are not missing genotype*diet interactions, and present the AL and CR data side by side in all relevant figures (e.g. Figure 3, Figure 3 —figure supplement 1, Figure 3 —figure supplement 2). This should be easy to do, though the updated analyses might lead to updated conclusions.

Thanks for the suggestion to perform statistical evaluation of main effects and interactions. The significant differences and trends are now reported throughout the figures in the main text and supplements. We have also provided the full statistical analyses in the Source data of Figure 2, Figure 3, Figure 3 —figure supplement 3, Figure 3 —figure supplement 4, Figure 4, Figure 4 —figure supplement 1, Figure 5 —figure supplement 2, Figure 6, Figure 6 —figure supplement 1.

4. Further analyze transcriptomic data for potential clues about the mechanisms underlying the hematological phenotypes (as they've done currently for the bone phenotype). Also highlight the limitation that the transcriptomic data are from distal tibia, whereas the bone analyses and hematological analyses are largely from other regions of the skeleton.

Thank you for encouraging us to dig deeper into the datasets. First, as shown in Figure 5 —figure supplement 2D, we observed that a gene cluster related to “Myeloid leukocyte differentiation” was largely upregulated in CR BMAd-Pnpla2^+/+^ mice; however, these increases were largely eliminated in CR BMAd-Pnpla2^-/-^ mice. Second, secretory protein analyses revealed that genes related to "hematopoietic stem cell proliferation" followed a similar pattern that was upregulated by CR in control mice, but not in CR BMAd-Pnpla2-deficient mice (Author response image 1) . Third, by comparing gene expression in BMAd-Pnpla2^+/+^ and BMAd-Pnpla2^-/-^ mice under AL and CR conditions, we found that the “hematopoietic progenitor cell differentiation pathway” was suppressed only in CR BMAd-Pnpla2^-/-^ mice, whereas expression of “leukocyte migration pathway” genes was increased in CR BMAd-Pnpla2^+/+^ mice, but not in CR BMAd-Pnpla2^-/-^ mice (New Figure 5 —figure supplement 3).

**Author response image 1. sa2fig1:** BMAd-Pnpla2 deficiency causes alterations in secretory protein gene expression. A. Differential genes regulated by CR in BMAd-Pnpla2^+/+^ mice were mapped with two independent secretome datasets: MetaZSecKB and VerSaDa. B. Overlapping genes between two secretome datasets were used for pathway analyses. . Z-Scores of genes enriched in hematopoietic stem cell proliferation pathway.

To address differences in where samples were taken for analyses, we have added to the limitation section of the discussion: “It should be noted that our RNAseq data was obtained from cBMAT of distal tibiae. It is likely that the fundamental interactions between BMAT and other cells within the marrow niche are similar between regions; however, we do not exclude the possibility that there may be unique interactions between rBMAT and osteoblast/hematopoietic cells.”

5. Update the OVX analyses to compare the OVX mice with ovary-intact females (as done for uterus masses in Figure 3 —figure supplement 4D). This would partially address the limitation, highlighted by Reviewer 2, that the authors haven't included appropriate sham or estrogen-replete controls for OVX mice. This limitation should be mentioned in the Discussion. Alternatively, the OVX experiment could be removed from the paper, as it is not particularly adding much.

Since there were already four groups in this cohort, it was challenging to add the ovary-intact females, which would have doubled the number of transgenic mice within the experiment. However, we had evaluated a sex- and age-matched female cohort and to provide context to our OVX data, we have now included these data as a control group in Figure 3 —figure supplement 4. Although these data are included as a reference, they are not included in the statistical analyses because they are from an independent cohort.

6. Need a proper description and discussion of the bone injury healing experiment.

We have modified the description of bone healing experiments with help from Dr. Kurt Hankenson, who has considerable experience in this area of research.

Reviewer #1 (Recommendations for the authors):Overall this is a good manuscript and no major revisions are recommended. However, there are several grammatical errors (e.g. line 521) that would require a careful proofreading to improve. Moreover, given the general readership of eLife, expansion of the introduction to include more background on topics that as written are fairly field-specific would certainly enhance the overall appeal to a wider audience.

We have gone through the whole manuscript carefully, and hope that the grammatical errors are corrected. Thank you for suggesting we add background for a broader audience. We have now provided an overview to adipocytes and bone marrow adipose tissue (BMAT) at the beginning of the introduction, as well as historical context underlying our current investigation.

Reviewer #2 (Recommendations for the authors):1) Consider the mechanism by which ATGL deficiency induces lipogenesis with respect to the following. There is a substantial accumulation of marrow fat in Pnpla2 -/- females under CR conditions which supersedes an amount of fat in control mice in the same conditions. This indicates massive lipogenesis rather than BMAd acting as a sink to circulating FA, as the authors propose.

We have now quantified the female data and included it in Figure 3 —figure supplement 3C. Male control mice have elevated BMAT that closely matches that in AL and CR BMAd-Pnpla2^-/-^ mice. However, as noted by the reviewer, the female CR BMAd-Pnpla2^-/-^ mice have more rBMAT than CR control mice, and more than AL BMAd-Pnpla2^-/-^ mice. Thus, BMAT increases with CR and/or pnpla2-deficiency in female mice. These observations highlight an underlying theme within this paper – that male and female mice differ in the physiology of the marrow niche. The basis for expansion of BMAT under these conditions is under active investigation within the field, including our lab. For increased number of BMAds, the mechanisms include increased adipogenesis and reduced dedifferentiation. For expanded size of adipocytes, the possibilities include increased de novo lipogenesis, increased uptake and storage of lipid, and impaired lipolysis. Although we did not profile gene expression in female mice, based on the RNAseq data from male mice, the adipogenesis/lipogenesis genes were dramatically increased in Pnpla2^+/+^ mice under CR conditions, whereas this effect was largely blocked in Pnpla2^-/-^ mice. Thus, it seems unlikely that the elevated number/size of adipocytes in CR BMAd-Pnpla2^-/-^ mice is due to increased storage of fat, but is instead due to an impairment of turnover. Dr. Philipp Scherer has shown quite convincingly that adipocytes of mammary gland and skin have a cycle of differentiation/dedifferentiation and as suggested in the original manuscript (Wang et al., 2018), this phenomenon may be also occurring in bone marrow given how the adipocytes ‘disappear’ with pregnancy and lactation (Honda et al., 2000), with GCSF administration (Li et al., 2019), and with hemolytic anemia (Weiss, 1965). However, lineage tracing of BMAds is challenging because of the high signal in stromal cells observed with inducible Adiponectin-based Cre or rtTA systems. Although these approaches label essentially 100% of adipocytes in white and brown depots, they are not completely penetrant to bone marrow adipose tissue, even with tamoxifen treatments over a long time (30 days). Thus, we will need to make technical breakthroughs before we can really understand the mechanism for expansion of BMAds under these conditions.

2) To assess whether estrogen protects female Pnpla2 -/- mice from negative effects of CR, mice were OVX. Surprisingly, OVX control and Pnpla2 -/- females "gained" bone when exposed to CR. There is no plausible explanation for this effect, which is in striking contrast to the CR effect in males. In addition, Sham-operated control is not provided in this experiment which poses the question of whether it is a bone gain or protection from bone loss. The conclusion that "metabolic benefits of CR diet combat detrimental effects of estrogen deficiency and/or aging" is overstated and applies only to females. This is an intriguing observation but needs more insight. What is its relevance to humans? Is there any evidence that malnourished postmenopausal women are gaining bone?

Thank you for raising this question and its translational significance. We initially were puzzled by our finding of increased trabecular bone during calorie restriction. But a closer look at the transcriptomics and a recent paper by one of our co-authors provides support for this novel observation. First, we noted that by pathway analysis skeletal system development was one of the top 3 gene networks up-regulated in the BMAd-Pnpla2^+/+^ mice but not the BMAd-Pnpla2^-/-^ mice undergoing 30% calorie restriction. Similarly extracellular matrix organization was also another up-regulated pathway. These responses were somewhat surprising since adipogenesis was the predominant cellular phenotype in the marrow. However, a closer look revealed that regulation of the cellular response to growth factor stimulus was another highly up-regulated network, implying there is an overall recruitment of progenitor cell populations in the niche, including pre-osteoblasts (Figure 5). In support of that tenet, unpublished studies from the Dr. Clifford Rosen’s laboratory have shown that marrow stromal cells from 30% calorie-deficient B6 male and female mice in vitro have enhanced numbers of CFU-Fs, and greater staining of alkaline phosphatase compared to ad libitum MSCs. This was further reinforced by our finding in males that markers of that population, Pth1r, Fgf2, Alpl and Cola1 were all up regulated with 30% CR in the BMAd-Pnpla2^+/+^ but not the BMAd-Pnpla2^-/-^ mice (Figure 5D-5F).

Our female study in Figure 3 —figure supplement 4 is complicated by the three variables we evaluated (OVX, diet and genotype). As suggested by reviewer 3, we went back and evaluated our data using two-way ANOVA, and in Figure 3 —figure supplement 3D, we observe that CR also increases bone volume fraction and decreases trabecular bone separation in ovary-intact female mice. Note that CR was for 12 weeks and was initiated when female mice were 16 weeks of age. We have reviewed previously the variable response to CR depending on mouse age and sex (Li and MacDougald, 2021).

In terms of the sham control groups, as mentioned above in the response to the editor: Since there were already four groups in this OVX cohort, it was challenging to add the ovary-intact females, which would have doubled the number of transgenic mice within the experiment. However, we had evaluated a sex- and age-matched female cohort and to provide context to our OVX data, we have now included these data as a control group in Figure 3 —figure supplement 4. Although these data are included as a reference, they are not included in the statistical analyses because they are from an independent cohort.

In respect to the translational significance of this work, recent findings from Fazeli et al. and Bredella et al. by our co-author Rosen (Bredella et al., 2022; Fazeli et al., 2021), demonstrated similar results to our mouse work. In that study of 23 human volunteers (mean age 33) who underwent first a high calorie diet (HCD) for ten days, and then after a two-week interval, 10 days of fasting, the increase in marrow adiposity was much more marked with fasting than with overnutrition, and male volunteers exhibited a much more consistent increase in BMAT than females. Moreover, when bone micro-architecture was measured by microCT at both the distal radius and tibia, trabecular bone volume fraction, trabecular thickness and trabecular plate thickness were all significantly increased with fasting compared to baseline measures (p<0.009). In contrast, no significant changes were noted with over-nutrition.

In response to the particular reviewer query, there is no evidence clinically that malnourished post-menopausal women gain bone with calorie restriction; indeed, over a longer period of time, it is likely that they could lose bone. For example, in the CALERIE study of human volunteers 25% calorie restriction over two years led to bone loss at three skeletal sites in both males and females (Villareal et al., 2016). Furthermore, Beavers et al. have shown in older women in the absence of estrogen that dietary restriction also causes bone loss (Wherry et al., 2021). Taken together our data suggest that short term calorie restriction in adult female mice increases trabecular bone mass due to an increase in recruitment of osteoblast progenitors that can make collagen. However, we postulate that longer term loss of energy (comparable to a human 10 day fast), as shown in the above noted human studies, would ultimately lead to bone loss, particularly in states of estrogen deficiency. The translational importance of our findings relates to the skeletal complications of long-term dietary manipulations. For example, intermittent fasting is now touted as a major weight-loss strategy, yet we know little about its impact on the skeleton, particularly in postmenopausal women. In fact, skeletal responses to intermittent fasting is the subject of a recently submitted R01 since it is clear that a better understanding of the molecular cellular and hormonal response in the marrow to dietary changes is needed.

3) Pnpla2 -/- male but not female mice have mildly affected myelopoiesis which is exacerbated in CR conditions. CR also affected hematopoietic marrow regeneration after irradiation which correlated with decreased number of hematopoietic stem cells. There is no evidence that these alterations are due to fuel deficiency in the marrow environment. The possibility that there are changes in the signaling milieu should be tested experimentally or considered based on transcriptomic data.

Thank you for raising this important caveat. We have added the following section to the discussion. “Although we have largely interpreted effects of Pnpla2 deficiency as causing a local energy shortfall, Pnpla2 deficiency may also alter downstream signaling activities. For example, ATGL deficiency prevents cAMP-dependent degradation of TXNIP and thus reduces glucose uptake and lactate secretion (Beg et al., 2021). ATGL-catalyzed lipolysis also provides essential mediators to increase activity of PGC-1α/PPARα and PPARd pathways (Haemmerle et al., 2011; Khan et al., 2015). Thus, it is conceivable that ATGL deficiency may alter the BMAd transcriptome and secretome to influence the marrow niche independent of its direct role as a fuel source.” Further we have added this sentence to the limitations section: “Although we have not provided direct evidence of BMAd-derived fatty acids fueling osteoblasts or hematopoietic cells, this line of investigation will require technical advances to distinguish BMAd-derived NEFA from those in circulation.”

We also appreciate your suggestion to analyze our RNAseq data to reveal possible mechanisms of impaired myelopoiesis. First, as shown in Figure 5 —figure supplement 2D, we found that gene expression related to “Myeloid leukocyte differentiation” was upregulated in CR BMAd-Pnpla2^+/+^ mice; however, these increases were largely eliminated in BMAd-Pnpla2^-/-^ mice. Second, interactions between BMAds and osteoblast/hematopoietic cells may be paracrine in nature; thus, we further mapped the RNAseq data with two independent secretome databases: MetaZSecKB (Meinken et al., 2015) and VerSaDa (Cortazar et al., 2017). Among the 1026 genes that were upregulated by CR in BMAd-Pnpla2^+/+^ mice, 88 were identified as highly likely secreted proteins in the MetaZSecKB database, whereas 69 genes encoding secreted proteins were identified by the VerSaDa database (Author response image 1) . Furthermore, we aligned the secretory proteins from these two independent databases and found 45 genes were shared, which were used for pathway analyses. Again, “extracellular matrix (ECM) organization” and “ECM-receptor interaction” were ranked as the top two pathways (Author response image 1). Interestingly, “hematopoietic stem cell proliferation pathway” was enriched for a number of genes, including Sfrp2, which was reported as an important factor for maintaining the HSC pool (Ruf et al., 2016); Thpo, which influences hematopoietic recovery after myeloablative stress (Gao et al., 2021); and Prg4, which is required for stimulating effects of parathyroid hormone on HSPCs (Novince et al., 2011). Z-Scores showed that Sfrp2, Thpo and Prg4 were upregulated in CR BMAd-Pnpla2^+/+^ mice, effects of which were largely eliminated by the deficiency of Pnpla2 in BMAd (Author response image 1). Third, further analyses of the differentially-regulated genes between CR BMAd-Pnpla2^+/+^ and CR BMAd-Pnpla2^-/-^ mice revealed that “hematopoietic progenitor cell differentiation pathway” was suppressed only in CR BMAd-Pnpla2^-/-^ mice, whereas “leukocyte migration pathway” was activated in CR BMAd-Pnpla2^+/+^ mice, but inhibited in CR BMAd-Pnpla2^-/-^ mice (New Figure 5 —figure supplement 3).

4) In regard to the bone regeneration model there is some misunderstanding. What the authors consider as trabecular bone is actually woven bone that is removed during the final stages of healing which occur at week 4 after injury. Referring to it as trabecular bone is not correct because the woven bone has a different origin and structure. In this model, day 9 after injury correlates with initiation of mineralization which robustness depends on previously formed hematoma and callus, and penetration of new vessels. It is possible that Pnpla2 deficiency affects early stages of bone healing not necessarily osteoblast bone-forming activity. Again, the transcriptomic data may give some clue, although they are done on marrow fat isolated from the distal tibia which is considered metabolically different than marrow fat in the proximal tibia where the bone defect was created.

We appreciate your professional insights and apologize for mis-characterizing the injury response. We have corrected the description of bone regeneration with the help from Dr. Kurt Hankenson, who is an expert in bone healing field. As shown in Figure 5C, genes in vasculature development, skeletal system development, extracellular matrix organization and ossification pathways are upregulated by CR in in BMAd-Pnpla2^+/+^ mice, but these pathways are inhibited in in CR BMAd-Pnpla2^-/-^ mice. The limitation of site-specific effects has been highlighted, although larger transgenic animals will be required to collect enough proximal tibial regulated BMAT to reveal the possible site-specific effects.

Reviewer #3 (Recommendations for the authors):General points:a. Adiponectin deficiency in the Pnpla2 WT vs KO mice:Please analyse circulating adiponectin in the Pnpla2 WT vs KO mice. This is important because you show that the BMAd-Cre mice have lower adiponectin and that the KO CR mice do not increase BMAT adiponectin expression to the same extent as the WT CR mice (i.e. Figure 5D and Figure 5 supplemental figure 2B) Notably, adiponectin KO mice have been shown to resist bone loss and BMAT expansion during CR (Zhu et al., 2020). Therefore, if the KO CR mice do have lower adiponectin than the WT CR mice then this might be influencing their bone phenotype. It would also be helpful to know if the BMAd-Cre mice do increase circulating adiponectin during CR, despite having lower adiponectin expression than the non-Cre mice.Because the hypoadiponectinaemia may be confounding, it would be helpful to compare the BMAd-Pnpla2+/+ and BMAd-Pnpla2-/- mice with Cre-negative Pnpla2-fl/fl mice, both under AL and CR conditions.

We have measured circulating adiponectin concentrations and reported the data in Figure 3 —figure supplement 1V. Although BMAd-Cre mice have hypoadiponectinemia, they still respond to caloric restriction by increasing circulating adiponectin concentrations. Circulating concentrations of adiponectin at baseline and with CR are similar in BMAd-Pnpla2^+/+^ and BMAd-Pnpla2^-/-^ mice. Because we noticed the BMAd-Cre, per se, caused hypoadiponectinemia, all animal used for experiments were BMAd-Cre positive mice, therefore unfortunately, we don’t have any non-Cre mice with CR treatment.

Related to these considerations, in lines 576-578 you mention a few previous studies of adiponectin and bone, but the Zhu 2020 paper about adiponectin KO and CR is not cited. Please update this part of the discussion to mention this very relevant paper.

Thanks a lot for the reference. It is cited in our manuscript.

b. Analysis of diet and diet-genotype effects:i. In Figure 3 and its supplemental figures 1 and 2 please can the data from ad libitum (AL) mice be analysed together with the data from the calorie-restricted (CR) mice? Currently the two diets are assessed separately and therefore it's not possible to detect effects of CR or if the BMAd Pnpla2 KO significantly alters these diet effects. Knowing this would be extremely informative as it may be that, while the KO alone has no effect within each diet, it might alter the CR effects. For some readouts this may also increase statistical power to detect significant effects of diet or genotype alone.

Our AL and CR cohorts were performed independently; we are cautious to directly combine them to present in the paper. As shown in Author response image 2, with the two-way ANOVA analysis after combining AL and CR cohorts together, the conclusions are not changed. We don’t observe an interaction between diet and genotype either.

**Author response image 2. sa2fig2:** BMAd lipolysis is required to maintain bone homeostasis in male mice under conditions of CR, but not when mice are fed ad libitum.

ii. Related to this, in Figure 3 (supplemental figures 3 and 4), Figure 4 (and its two supplemental figures), Figure 5 (supplemental figure 2B) and Figure 6 the four genotype-diet groups are analysed side-by-side, but they are compared using 1-way ANOVA. For these figures using a 2-way (or sometimes 3-way) ANOVA is more appropriate and, as above, is required if you are to detect diet effects and genotype-diet interactions. For example, the time course data in Figure 4 supplemental figure 1C-E should be assessed by 3-way ANOVA with genotype, diet and time as the independent variables, while the data in Figure 4 and panels F-G of the supplemental figure should be assessed by 2-way ANOVA with genotype and diet as the independent variables. As above, this is important because you can then test for interactions between the variables, which might reveal KO effects that cannot be detected by 1-way ANOVA. In some cases, the 2-way ANOVA is also required to support existing statements in the results. For example, Line 303 ("…mature neutrophils are affected only by CR (Figure 4H)") and Lines 307-308 ("fewer granulocyte progenitor colonies (CFU-G) from CR BMAd-Pnpla2-/- mice (Figure 4I)") are not supported by any statistical analysis.

Original data of Figure 3 (and its supplemental figures 3 and 4), Figure 4 (and its supplemental figure 1), Figure 5 (and its supplemental figure 2B) and Figure 6 (and its supplemental figure 1), all the four genotype-diet groups have been analyzed with two-way or three-way ANOVA accordingly. Conclusions have been updated and the analytic data are included in the source data excel file. To address the two specific contexts you mention: we now have included the statement that “The recovery of neutrophils, preneutrophils, immature and mature neutrophils following irradiation was impaired by CR, and Pnpla2-deficiency further reduced ability of neutrophils and mature neutrophils to regenerate (Figure 4 E-4H).” For Figure 4I, no statistical differences were observed, thus we softened our statement for CR BMAd-Pnpla2 -/- mice to “a trend of fewer…”.

c. In the Results section the findings frequently are described in the present tense. Convention is to report the results in the past tense (i.e. what was found) and then, in the Discussion, to interpret these data in the present tense (i.e. what the results show/reveal etc). Please can this be updated throughout the Results section?

We appreciate the reviewer’s suggestions for writing. The tenses have been changed throughout the manuscript.

d. Genotypes of BMAd-Cre mice:i. In Supplemental Figure 2, you distinguish homozygous vs heterozygous BMAd-Cre mice. I think it would help the reader to clarify that this relates to the FAC transgene, but it's not clear if these mice have one or two alleles of the Osterix-FLPo transgene. Please can you clarify in the text of the results.

We don’t observe any differences in the FLPo efficiency between one or two alleles. We also did not notice any metabolic or bone growth phenotype (Author response image 3) . Thus, cohorts were generally homozygous for Osterix-FLPo with a minority of mice being heterozygous.

**Author response image 3. sa2fig3:** Phenotypes of Osterix-FLPo wild type, heterozygous and homozygous mice. Osterix-FLPo mice were euthanized at 24 weeks of age. GTT was performed at 22 weeks of age (A). Body weights, tissue weights and bone lengths were measured during dissection (B-I). Tibiae were collected for mCT analyses; trabecular (J-O) and cortical bone (P-Q) parameters were measured.

ii. Lines 174-175: You state that mice used for the Pnpla2 KO studies were positive for both Osterix-FLPo and FAC; were they homozygous for both transgenes or was one of the heterozygous (etc)? This needs to be clarified in the text.

We have now stated that “To minimize variability between treatments, all mice used in the following studies were positive for both Osterix-FLPo (predominately two alleles) and FAC (mostly two alleles).

Specific points:e. Figure 1 —figure supplement 1:i. For 1C I'm confused by the genotype labelling for the Osterix mouse above the blot: does the '+' refer to the presence of FLPo? It's confusing as usually '+' would refer to the WT allele; please clarify in the figure and the legend.

Apologies for the confusion. “+” indicates the presence of FLPo or FAC (insertion), which has been clarified in the legend.

ii. In the main text (lines 126-128) you state, "Cre in the correct orientation… is only observed in caudal vertebral DNA of mice positive for at least one copy each of Osterix (Mut band) and FAC (Ori band)". This is shown in Supp Figure 1F so please refer to this figure after making this statement in the text. Also, it is logical to move Supp Figure 1F to Supp Figure 1D, as it follows immediately from 1C.

We have referred Supp Figure 1F in the text and adjusted the order of panels according to reviewer’s suggestion

f. Lines 144-145, "Indeed, Cre efficiency is ~80% in both male and female mice over 16 weeks of age with one Cre allele, and over 90% at 12 weeks of age in mice with two Cre alleles". Please explain how you calculated these percentages of efficiency; no quantification is shown in the figures.

We have now rigorously quantified the data based on red versus green BMAds in BMAd-mTmG mice. The EGFP+ cell percentages are included in Figure 1—figure supplement 1I.

g. Lines 160-161: Please can you reword this text for clarity, e.g. "We found that insertion of the IRES-Cre cassette caused a small, non-significant decrease in Adipoq mRNA expression in whole caudal vertebrae or distal tibiae (Figure Supp 2A), whereas Adipoq mRNA in WAT…"

Reworded as reviewer suggested.

h. Line 166: The greater decrease in circulating adiponectin is clear from the data. Therefore, please revise the wording to "Circulating adiponectin concentrations were decreased even further…".

Reworded as reviewer suggested.

i. Figure 1 —figure supplement 2F: please can you refer to this in the figure legend, e.g. "(F) High, medium and low molecular weight forms of adiponectin were quantified by Image J."

Thanks for the careful reading. “(F)” is added.

j. Figure 2i. 2B: There is still some ATGL fluorescence in the proximal tibia of the KO mice. Where is this coming from? Perhaps ATGL is expressed earlier than adiponectin in BMAd development, so some differentiating BMAds may still have intact ATGL expression?

Sorry for the dirty background. The remaining fluorescence is mostly between cells, which may be due to the insufficient washing after staining.

ii. 2D-F: The differences look greatest in the proximal tibial diaphysis, above the tibia-fibula junction. Is there a reason that this region was not quantified by osmium?

The region between growth plate and tibia/fibula junction was quantified as shown in Author response image 4. It was not included in our figure because we ran out of space to include it. Plus, it is consistent with proximal metaphysis data and does not affect the conclusion.

**Author response image 4. sa2fig4:** Quantification of rBMAT in tibial diaphysis region.

k. Lines 220-222, "These data also indicate that expansion of BMAT is not sufficient to cause bone loss and that the correlation between elevated fracture risk and expansion of BMAT under a variety of clinical situations is not necessarily a causal relationship." This is an interesting and important point but is better suited to the Discussion section.

We moved this sentence into discussion.

l. Figure 3 (main figure and Supplemental figures):i. Figure 3 Supp 1B vs 1K: Was glucose tolerance improved by CR? Fasting and peak glucose in the CR mice (1K) look similar to in the ad libitum mice (1B). It would be informative to calculate the AUC for the GTTs and to then plot this for all four groups on the same axis, comparing by 2-way ANOVA.

Because the AL and CR cohorts were independent, we are cautious to combine them together. In addition, they were lean mice with regular chow diet. The improvement of glucose metabolism may not be obvious at this case. Therefore, the AUC was calculated separately.

ii. In Figure 3 Supp 1O you show nicely that total BMAT volume is similar between KO and control during CR. Is BMAd size the same? i.e. do the KO mice still have larger BMAds than control mice during CR?

The representative images and quantification of adipocyte sizes are now included in Figure 3 —figure supplement 1Q-1S. “Although Pnpla2^-/-^ mice fed ad libitum had increased rBMAT compared to controls (Figure 3 —figure supplement 1F and 1G), CR stimulated rBMAT expansion in control mice such that BMAT volume and BMAd size were comparable between genotypes (Figure 3 —figure supplement 1O-1S).”

iii. Lines 243-244: You state "we observed a trend (P = 0.07) towards reduced NEFA concentrations in bone marrow supernatant of CR BMAd-Pnpla2-/- mice". However, the same trend seems to occur for the AL mice (Supplemental 3I). Please can you add P values to Figure 3 Supp 1I so that the AL data can be compared more directly with glycerol and NEFA during CR (Figure 3 Supp 1R)? Better still would be to have all four groups compared on a single axis (as suggested above in General Point b), as this may give you sufficient power to detect a significant genotype effect on NEFA in the BM supernatant.

The P value for Figure 3 Supp 1I has been added. Again, we are cautious to put independent cohorts together to do the comparison.

iv. In line 255 you state that TRACP5b is decreased but from the figure this isn't significant, so please temper this statement in the main text. Alternatively, if you were to analyse the AL and CR mice together by 2-way ANOVA you might then have enough power to detect a significant decrease in the KO mice, irrespective of diet.

We have tempered this statement by saying “or showed a strong trend towards reduction…”

m. Was DAPI used for live/dead staining in your flow cytometry? This is not absolutely essential but should be stated in the Methods section if this was or was not done.

We did not use DAPI. Instead, dead cells and doublets were excluded based on FSC and SSC distribution.

n. Table 2: Please confirm that these P values are adjusted for multiple comparisons and update the Table legend to state how statistical significance was determined.

Multiple comparisons with FDR methods have been performed, adjusted P values are included in Tables 1 and 2.

o. Figure 3 —figure supplement 3:i. BM adiposity is shown histologically in panel C but this is not quantified, so it's not clear to what extent this is altered by genotype and/or diet. Please can you quantify BM adiposity from these images? Even better would be to use osmium staining so that the female data could be compared with the male data in Figure 3 supplement 1.

Unfortunately, we did not perform osmium staining in this cohort. But we quantified the percentages of BMAT area in proximal tibial, please find the new Figure 3 —figure supplement 3C (right).

p. In Figure 5B-C and the related supplemental figures, what's the rationale for comparing CR KO mice to AL WT mice? I think this comparison is invalid because it is being made across two independent variables. For example, if the target gene expression is similar in CR WT vs CR KO, but (e.g.) lower in AL WT vs AL KO, then the fold change vs AL WT will be similar for CR WT and CR KO, but the fold change for CR KO vs AL KO will be greater. To my mind, this would mean that the KO is altering the effect of CR, but your approach would not detect this. So, if the goal is to identify genes regulated by CR irrespective of Pnpla2 deficiency then this should be done by comparing the effects of CR within each genotype.

We tried to compare with the same control to make sure the baseline is the same. As Reviewer noticed, the data set is very similar regardless of the control groups (Author response image 5) . We have corrected the comparison with AL KO, which does not change the following conclusion though.

**Author response image 5. sa2fig5:** Comparison of gene set with different control groups.

q. The finding of impaired bone regeneration with CR and Pnpla2 KO is consistent with your conclusion, i.e. that BMAd lipolysis is important for bone regeneration. However, another interpretation relates to the findings of Ambrosi 2017, who show that BMAds impair fracture healing (Ambrosi et al., 2017). Their data suggest that BMAds can secrete DPP4 and that this impairs bone healing. Therefore, another possibility for your observation is that increased BMAT in the Pnpla2 KO mice results in increased production of DPP4 in the bone marrow, which impairs bone regeneration. Please can you measure DPP4 expression within the BM and/or discuss this possibility in the Discussion section?

This is an interesting point. However, we don’t find differences in Dpp4 expression between BMAd-Pnpla2^+/+^ and BMAd-Pnpla2^-/-^ mice, although it is significantly inhibited by CR (Author response image 6) .

**Author response image 6. sa2fig6:** Z-scores of Dpp4 expression in BMAT in response to CR and Pnpla2 deficiency.

r. Line 453: Please correct 'Figure 4I' to 'Figure 4H'. Also, you can't really state that BMD declines when this effect is not statistically significant; can you test this further (e.g. use 2-way ANOVA and/or in a second cohort of mice), or at least temper the language used to describe this effect?

Figure 6I is corrected into 6H. We tempered the tone of our statement to “trabecular bone volume fraction declined with cold exposure in BMAd-Pnpla2^-/-^ mice and bone mineral density showed a similar trend (Figure 6H).”

s. You show that myelopoiesis is impaired by BMAd-specific Pnpla2 deficiency, but do not go into the mechanism about why only the myeloid system, and not the CLP or HSC (which are also in the bone marrow), is dependent on energy supply from BMAds under CR. Please can you speculate on this in the Discussion?

There is complementary interplay between epigenetic regulation and fatty acid metabolism that is mediated by exogenous lipid oxidation directly altering methylation states and by the provision of acetyl-CoA for acetylation (Ivashkiv and Park, 2016). The mechanisms by which adaptation to altered nutrient availability may specifically regulates development of myeloid lineages is a fascinating subject; however, it’s beyond the purview of this manuscript.

References

Beg, M., Zhang, W., McCourt, A.C., and Enerback, S. (2021). ATGL activity regulates GLUT1-mediated glucose uptake and lactate production via TXNIP stability in adipocytes. J Biol Chem 296, 100332.

Bredella, M.A., Fazeli, P.K., Bourassa, J., Rosen, C.J., Bouxsein, M.L., Klibanski, A., and Miller, K.K. (2022). The effect of short-term high-caloric feeding and fasting on bone microarchitecture. Bone 154, 116214.

Cortazar, A.R., Oguiza, J.A., Aransay, A.M., and Lavin, J.L. (2017). VerSeDa: vertebrate secretome database. Database (Oxford) 2017.

Fazeli, P.K., Bredella, M.A., Pachon-Pena, G., Zhao, W., Zhang, X., Faje, A.T., Resulaj, M., Polineni, S.P., Holmes, T.M., Lee, H., et al. (2021). The dynamics of human bone marrow adipose tissue in response to feeding and fasting. JCI Insight 6.

Gao, L., Decker, M., Chen, H., and Ding, L. (2021). Thrombopoietin from hepatocytes promotes hematopoietic stem cell regeneration after myeloablation. Elife 10.

Haemmerle, G., Moustafa, T., Woelkart, G., Buttner, S., Schmidt, A., van de Weijer, T., Hesselink, M., Jaeger, D., Kienesberger, P.C., Zierler, K., et al. (2011). ATGL-mediated fat catabolism regulates cardiac mitochondrial function via PPAR-alpha and PGC-1. Nat Med 17, 1076-1085.

Honda, A., Kurabayashi, T., Yahata, T., Tomita, M., Matsushita, H., Takakuwa, K., and Tanaka, K. (2000). Effects of pregnancy and lactation on trabecular bone and marrow adipocytes in rats. Calcif Tissue Int 67, 367-372.

Ivashkiv, L.B., and Park, S.H. (2016). Epigenetic Regulation of Myeloid Cells. Microbiol Spectr 4.

Khan, S.A., Sathyanarayan, A., Mashek, M.T., Ong, K.T., Wollaston-Hayden, E.E., and Mashek, D.G. (2015). ATGL-catalyzed lipolysis regulates SIRT1 to control PGC-1alpha/PPAR-alpha signaling. Diabetes 64, 418-426.

Li, Z., Hardij, J., Evers, S.S., Hutch, C.R., Choi, S.M., Shao, Y., Learman, B.S., Lewis, K.T., Schill, R.L., Mori, H., et al. (2019). G-CSF partially mediates effects of sleeve gastrectomy on the bone marrow niche. J Clin Invest 129, 2404-2416.

Li, Z., and MacDougald, O.A. (2021). Preclinical models for investigating how bone marrow adipocytes influence bone and hematopoietic cellularity. Best Pract Res Clin Endocrinol Metab, 101547.

Meinken, J., Walker, G., Cooper, C.R., and Min, X.J. (2015). MetazSecKB: the human and animal secretome and subcellular proteome knowledgebase. Database (Oxford) 2015.

Novince, C.M., Koh, A.J., Michalski, M.N., Marchesan, J.T., Wang, J., Jung, Y., Berry, J.E., Eber, M.R., Rosol, T.J., Taichman, R.S., et al. (2011). Proteoglycan 4, a novel immunomodulatory factor, regulates parathyroid hormone actions on hematopoietic cells. Am J Pathol 179, 2431-2442.

Ruf, F., Schreck, C., Wagner, A., Grziwok, S., Pagel, C., Romero, S., Kieslinger, M., Shimono, A., Peschel, C., Gotze, K.S., et al. (2016). Loss of Sfrp2 in the Niche Amplifies Stress-Induced Cellular Responses, and Impairs the In Vivo Regeneration of the Hematopoietic Stem Cell Pool. Stem Cells 34, 2381-2392.

Villareal, D.T., Fontana, L., Das, S.K., Redman, L., Smith, S.R., Saltzman, E., Bales, C., Rochon, J., Pieper, C., Huang, M., et al. (2016). Effect of Two-Year Caloric Restriction on Bone Metabolism and Bone Mineral Density in Non-Obese Younger Adults: A Randomized Clinical Trial. J Bone Miner Res 31, 40-51.

Wang, Q.A., Song, A., Chen, W., Schwalie, P.C., Zhang, F., Vishvanath, L., Jiang, L., Ye, R., Shao, M., Tao, C., et al. (2018). Reversible De-differentiation of Mature White Adipocytes into Preadipocyte-like Precursors during Lactation. Cell Metab 28, 282-288 e283.

Weiss, L. (1965). The structure of bone marrow. Functional interrelationships of vascular and hematopoietic compartments in experimental hemolytic anemia: an electron microscopic study. J Morphol 117, 467-537.

Wherry, S.J., Miller, R.M., Jeong, S.H., and Beavers, K.M. (2021). The Ability of Exercise to Mitigate Caloric Restriction-Induced Bone Loss in Older Adults: A Structured Review of RCTs and Narrative Review of Exercise-Induced Changes in Bone Biomarkers. Nutrients 13.